



# MUNICH v2.0: A street-network model coupled with SSH-aerosol (v1.2) for multi-pollutant modelling

Youngseob Kim[1], Lya Lugon[1,†], Alice Maison[1,2], Thibaud Sarica[1], Yelva Roustan[1], Myrto Valari[3], Yang Zhang[4], Michel André[5], and Karine Sartelet[1]

[1]CEREA, École des Ponts, EDF R&D, Marne-la-Vallée, France
[2]Université Paris-Saclay, INRAE, AgroParisTech, UMR EcoSys, Thiverval-Grignon, France
[3]Laboratoire de Météorologie Dynamique, Sorbonne Université, École Polytechnique, IPSL, École Normale Supérieure, CNRS, Paris, France
[4]Department of Civil and Environmental Engineering, Northeastern University, Boston, MA, USA
[5]Department COSYS, Université Gustave Eiffel, Bron, France
[†]Now at Max-Planck-Institut for Meteorology, Hamburg, Germany

**Correspondence:** Youngseob Kim (youngseob.kim@enpc.fr), Karine Sartelet (karine.sartelet@enpc.fr)

**Abstract.** A new version of the street-network model, Model of Urban Network of Intersecting Canyons and Highways version 2.0 (MUNICH v2.0) is presented. The comprehensive aerosol model SSH-aerosol is implemented in MUNICH v2.0 to simulate the street concentrations of multi pollutants including secondary aerosols. The implementation uses the Application Programming Interface (API) technology so that the SSH-aerosol version may be easily updated. New parameterisations are

also introduced in MUNICH v2.0, including a non-stationary approach to model reactive pollutants, particle deposition and resuspension, and a parameterisation of the wind at roof level. A test case over a Paris suburb is presented for model evaluation and illustration of the impact of the new functionalities. The implementation of SSH-aerosol leads to an increase of 11% of $PM_{10}$ concentration, because of secondary aerosol formation. Using the non-stationary approach rather than the stationary one leads to a decrease in $NO_2$ concentration by 16%. The impact of particle deposition and resuspension on pollutant

concentrations in the street canyons is low.

## 1 Introduction

More than half of the population now lives in urban areas (Ritchie and Roser, 2018), and is often exposed to high concentrations of nitrogen dioxide ($NO_2$) and fine particulate matter (PM) of diameters lower than and equal to 2.5 µm ($PM_{2.5}$, Krzyzanowski et al., 2014). In many European cities, there are densely built districts with street-canyon configurations. Street-level air quality

has been reported to be worse than that in the surrounding area because of the presence of air pollutant sources. In particular, high concentrations of $NO_2$ (Cyrys et al., 2012), black carbon (Putaud et al., 2010; Lugon et al., 2021b), and organics have been reported (Putaud et al., 2010; Airparif, 2011).

Air quality models provide a useful tool to understand the phenomena of pollution in street canyons (Lugon et al., 2021b) and to estimate the impact of emission scenarios to reduce pollution. Different types of models may be used to represent the

pollution in street canyons. Computational fluid dynamics (CFD) models describe finely the urban geometry, the air flow and





the pollutant concentrations, e.g., Code_Saturne (Milliez and Carissimo, 2007; Thouron et al., 2019), OpenFOAM (Jeanjean et al., 2015; Wu et al., 2021), STAR-CCM+ (Santiago et al., 2017), the PALM model (Wolf et al., 2020; Zhang et al., 2021). However, the computational cost is too high for operational purpose to predict the pollutant concentrations if applied to a city district with a large street network (at least hundreds street segments) (Vardoulakis et al., 2003). Parametric models are another type. They are suitable for operational purpose because of low computational cost. Some parametric models are based on a Gaussian dispersion methodology to represent emitted traffic-related pollutants, such as a Gaussian plume or puff, e.g., Polyphemus (Briant et al., 2013), CALINE4 (Benson, 1992), etc. Because they can not represent a street-canyon configuration, they are modified to include a specific module to represent this particular geometry, e.g. OSPM (Berkowicz, 2000), SBLINE (Namdeo and Colls, 1996), ADMS-Urban (McHugh et al., 1997). Other parametric models use parameterisations based on CFD modelling or wind-tunnel experiments to describe the flow in each street and the exchange from street to street, and between streets and the overlying atmosphere. The transport of pollutants from one street to another is taken into account through intersections, e.g. SIRANE (Soulhac et al., 2011) and the Model of Urban Network of Intersecting Canyons and Highways MUNICH (Kim et al., 2018). The flow above the street network is represented by a Gaussian dispersion methodology (SIRANE), or by one or two-way nesting to a regional model (MUNICH).

The streets are discretized with an Eulerian approach and boxes representing the street-segment volumes. Breaking away from the Gaussian methodology, this approach allows to model the reactivity of pollutants as they are transported from the regional scale (background concentrations) to the street. Lugon et al. (2020) showed that it is crucial to couple the transport of pollutants in the street and chemistry finely, using a non-stationary approach avoiding a steady-state assumption, in order to represent the concentrations of reactive pollutants, such as $NO_2$. By coupling MUNICH to the aerosol model SSH-aerosol (Sartelet et al., 2020), Lugon et al. (2021a) showed that the formation of secondary aerosols is important not only at the regional scale, but also at the street level. This paper presents the version 2.0 of MUNICH. The different model improvements of Lugon et al. (2020, 2021a) have been implemented, as well as the modelling of deposition and resuspension of Lugon et al. (2021b). The coupling to the model SSH-aerosol has also been improved and automated. New parameterisations of the flow in the street are also added. A reference test case is presented for model evaluation and to illustrate the behaviour and the capabilities of MUNICH.

The description of the model along with major updates from v1.0 to v2.0 is summarized in section 2. Section 3 presents the simulation domain and the set up of the reference test case, which is compared to observations of $NO_2$, nitric oxide (NO), $PM_{2.5}$ and particulate matter of diameters lower than $10\,\mu m$ ($PM_{10}$). In sections 4, 5 and 6, different sensitivity simulations are presented to understand how these updates influence the street concentrations. They are classified depending on whether they concern transport (section 4), chemistry (section 5) and deposition/resuspension (section 6). Finally, two other sensitivity simulations on important parameters for the modelling and applications of MUNICH are performed (influence of building aspect ratio and effects of removing car traffic from specific streets).





## 2 Description of the model and major updates

The version 1.0 of MUNICH is described in Kim et al. (2018). Only the main concepts are reviewed here. In MUNICH, a street network is divided into street segments and intersections. A street segment is bounded by intersections with other street segments. A street segment is represented by one cuboid-type box and concentrations are assumed to be homogeneous in the corresponding volume, which is estimated as the product of segment length, width and average building height. The fluxes of pollutants emitted in a street segment, from human activities to natural sources (e.g. trees) are diluted in this volume. Pollutant concentrations are only evaluated in each street segment and not at intersections, which are defined to represent the street-to-street advective transfer of pollutants and part of the exchanges with the overlying atmosphere (Soulhac et al., 2009). The exchanges from a street segment to the overlying atmosphere are also computed at the top of each street segment. If the formulation of Salizzoni et al. (2009) is used, they depend on the standard deviation of the vertical wind velocity at roof level, which depends on atmospheric stability, and on the concentration gradients between the street and above. As detailed in MUNICH v1.0 (Kim et al., 2018), this formulation may be modified to take into account the influence of the street ratio ($H/W$), as suggested by Schulte et al. (2015), and detailed in Appendix B.

Pollutants are also advected from street to street after averaging the vertical wind profile in the street. This profile depends on the wind velocity at the roof level, which depends itself on the meteorological data above the streets (e.g., wind speed and direction) and the street segment characteristics (e.g., street segment direction, street width, building height). As detailed in MUNICH v1.0, two formulations may be used to represent the wind profile within the streets: the exponential formulation (Lemonsu et al., 2004) or the analytical formulation from SIRANE (Soulhac et al., 2008). These formulations depend on the wind velocity at the roof level $u_H$ (see Appendix B), for which a new formulation is proposed here (see section 2.2 based on the work of Macdonald et al. (1998)).

Emission may be both in the gas and particle phases. The chemical transformations of the pollutants are modelled by a chemical kinetic mechanism for the gas-phase and/or by an aerosol model representing the aerosol dynamics (nucleation, condensation/evaporation and coagulation) and the transfers between the gas and particle phases.

The loss fluxes due to deposition are represented through parameterisations of dry deposition and wet scavenging. An approach to estimate resuspension is added in MUNICH v2.0, following Lugon et al. (2021b).

Many modelling options are included to represent the different physico-chemical processes taken into account in MUNICH. They are presented in Appendix B. An academic test case is set up in this section to illustrate how the pollutants are transported within the street network. Then, the section describes the new features in comparison to MUNICH v1.0.

### 2.1 Advection through intersections

At intersections, the pollutant mass flux from one street to others can be computed by estimating the balance of the air-volume fluxes among the streets that are connected to the intersection.

A simplified street network with 12 street segments was designed to perform an theoretical test case and illustrate how mass fluxes are modelled. For simplicity, the wind speed at roof top is fixed to an arbitrary value (5 or 10 m s$^{-1}$), and a pollutant is



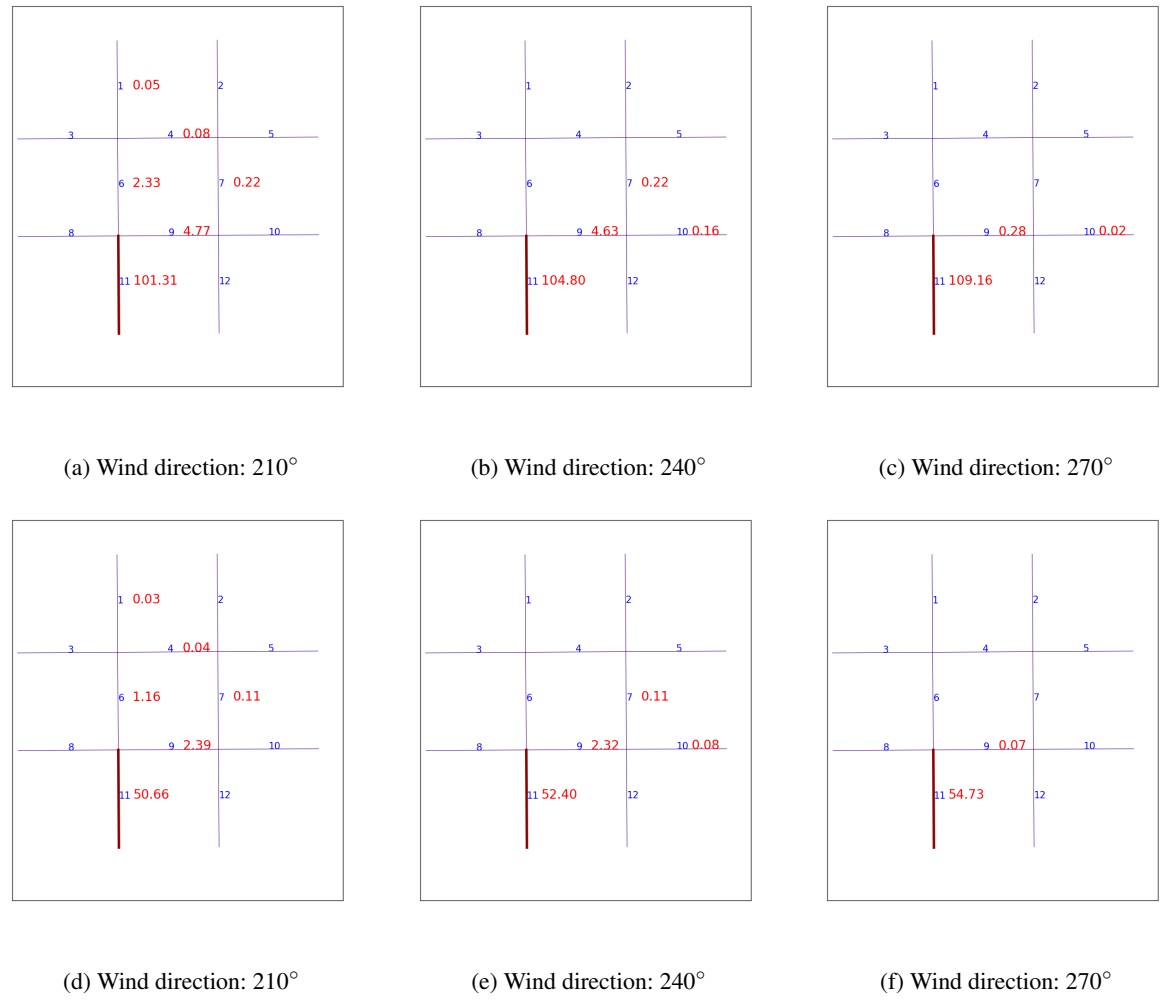

**Figure 1.** Variation of pollutant concentrations in a street network depending on wind direction. The wind speed is $5\,\mathrm{m\,s}^{-1}$ for the upper three cases and $10\,\mathrm{m\,s}^{-1}$ for the lower three cases. The wind direction is given from the North (top of the figure). The blue numbers are the street ID and the red numbers are the concentrations in $\mu\mathrm{g\,m}^{-3}$. Pollutant is emitted only in the street segment 11.

emitted in only one single street segment (number 11 in Figure 1). Figure 1 displays the mass concentration calculated using MUNICH in the different street segments (red numbers). The concentrations are the highest in the street segment where the pollutant is emitted. The concentrations vary depending on the wind direction above the streets. For higher wind speed at the roof level ($10\,\mathrm{m\,s}^{-1}$, see Figures 1d-f), the pollutant concentrations are lower over all street segments. This is due to an increase

90    of the advection and also an increase of the vertical transfer by turbulence at rooftop.



## 2.2 Wind velocity at the roof level

The computation of the vertical wind profile within the street depends on the wind velocity determined at the roof level $u_H$ (see equations B12 and B13 of Appendix B). $u_H$ was computed following Soulhac et al. (2008) in MUNICH v1.0, based on a 2D parameterisation of the wind field along the street axis. Now it may be computed depending on the street characteristics using a logarithmic wind profile above the buildings, as defined in Macdonald et al. (1998). This wind profile corresponds to an average profile over a relatively large urban scale, such as a relatively homogeneous district or city. It is based on the calculation of a displacement height ($d_c$) and a roughness length ($z_{0c}$) for the homogeneous urban canopy area (district) considered. Note that the roughness length of the district is typically on the order of magnitude of $1\,\text{m}$ (see Figure 2) when those of street walls and road pavement are on the order of $1\,\text{mm}$.

$$\frac{d_c}{\overline{H}} = 1 + \Delta^{-\lambda_P} \left( \lambda_P - 1 \right) \tag{1}$$

$$\frac{z_{0c}}{\overline{H}} = \left( 1 - \frac{d_c}{\overline{H}} \right) \exp\left( - \left( 0.5\delta \frac{C_{D_b}}{\kappa^2} \left( 1 - \frac{d_c}{\overline{H}} \right) \lambda_F \right)^{-0.5} \right) \tag{2}$$

where $\Delta$ and $\delta$ are empirical constants ($\delta$ = 1.0 and $\Delta$ = 4.43 for staggered arrays ; $\delta$ = 0.55 and $\Delta$ = 3.59 for square arrays; Macdonald et al. 1998, the values for staggered arrays are used in MUNICH v2.0), $C_{D_b}$ is the building drag coefficient usually equal to 1.2 (Macdonald et al., 1998), $\kappa$ is the Von Kármán constant ($\kappa$ = 0.41).

$\lambda_P$ and $\lambda_F$ are respectively the plan and frontal area densities of obstacles calculated as:

$$\lambda_P = \frac{A_P}{A_T} = \frac{\overline{W_{building}}\,\overline{L}}{\left(\overline{W_{building}} + \overline{W_{street}}\right)\overline{L}} \;\; \text{and} \;\; \lambda_F = \frac{A_F}{A_T} = \frac{\overline{H}\,\overline{L}}{\left(\overline{W_{building}} + \overline{W_{street}}\right)\overline{L}} \tag{3}$$

$A_F$, $A_P$ and $A_T$ are respectively the frontal, plan and lot area of obstacles ($A_T$ corresponds to the total area divided by the number of obstacles). Those surface ratios are calculated from the average characteristics of the streets in the considered district: building height ($\overline{H}$), street width ($\overline{W_{street}}$), building width ($\overline{W_{building}}$) and street length ($\overline{L}$) that cancels in the both equations.

Finally, $u_H$ is calculated for each street of building height $H$ as:

$$u_H = \frac{u_*}{\kappa} \ln\left( \frac{H - d_c}{z_{0c}} \right) = u_{ref} \times \frac{\ln\left( \frac{H - d_c}{z_{0c}} \right)}{\ln\left( \frac{z_{ref} - d_c}{z_{0c}} \right)}. \tag{4}$$

Depending on the chosen input parameters, $u_H$ can be calculated from the friction velocity $u_*$ (in $\text{m.s}^{-1}$) defined at urban canopy scale or from a wind speed at a reference altitude above the street ($u(z_{ref}) = u_{ref}$ in $\text{m.s}^{-1}$). For each street, only the axial component of $u_H$ is considered to compute the average wind speed in the street direction. Therefore, the horizontal transport of pollutants in the street depends on the angle between the wind direction and the street orientation.



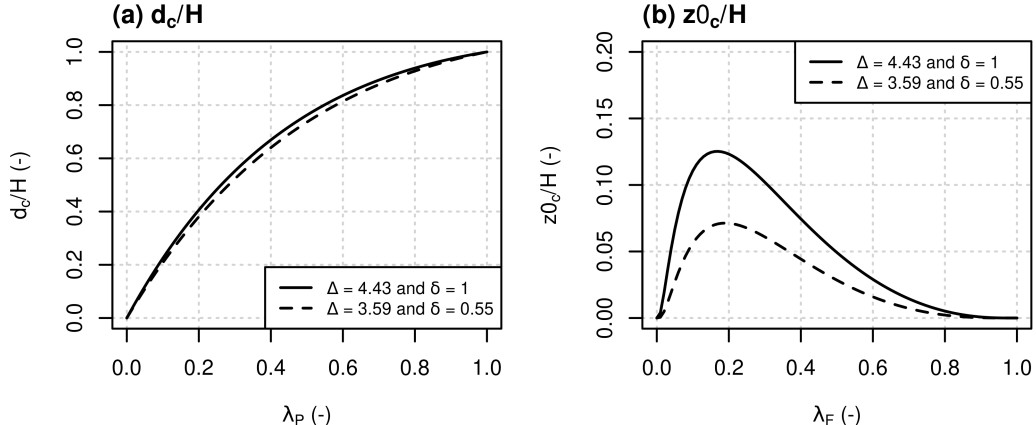

**Figure 2.** $d_c/\overline{H}$ and $z_{0_c}/\overline{H}$ as a function of the plan and frontal area densities calculated with Macdonald et al. (1998) equations.

## 2.3 Concentrations of reactive species: non-stationary approach

In MUNICH v1.0, to calculate the concentrations in a street segment, a first-order splitting scheme between "transport" (including removal processes) and chemistry is used, with fixed splitting time steps (typically 100 s). This numerical approach holds for slowly-reacting species, but it fails to represent the time evolution of fast-reacting species. The characteristic time

scales of fast chemistry processes may be similar (or faster) than those of transport in and out the street. A new algorithm is presented in Lugon et al. (2020) to overpass the steady-state assumption for transport (i.e. the stationary approach). At the first time iteration, the characteristic time scale of transport is estimated, then transport and chemistry are solved sequentially on a time step corresponding to this characteristic time. Transport is solved using an explicit two-stage Runge-Kutta algorithm (explicit trapezoidal rule of order 2) or a semi-implicit Rosenbrock algorithm, and chemistry is solved with smaller time steps

using a Rosenbrock algorithm or the solver used in SSH-aerosol (two-stage Runge-Kutta or two-step algorithms). Chemistry (gas-phase chemistry and aerosol dynamics) are solved using sub-time steps, because they correspond to a stiff set of equations with very fast processes such as radical chemistry. The time step is adapted depending on the evolution of the concentrations due to transport-related processes.

Using a beta version of MUNICH, Lugon et al. (2020) showed that this algorithm is numerically stable for reactive species,

unlike the one using the stationary assumption. The effects of this new algorithm in MUNICH v2.0 are presented in section 5.2.

## 2.4 Coupling to SSH-aerosol (v1.2)

The chemical composition of particles in streets differs from those above, mostly because of emitted pollutants within the street, for example from traffic (Lugon et al., 2021a). Within the streets, emitted pollutants mix with those from above the street and undergo chemistry. In MUNICH v1.0, only gas-phase chemistry is taken into account, and the CB05 chemical

kinetic mechanism (Yarwood et al., 2005) is implemented to simulate the gas-phase concentrations (Kim et al., 2018).





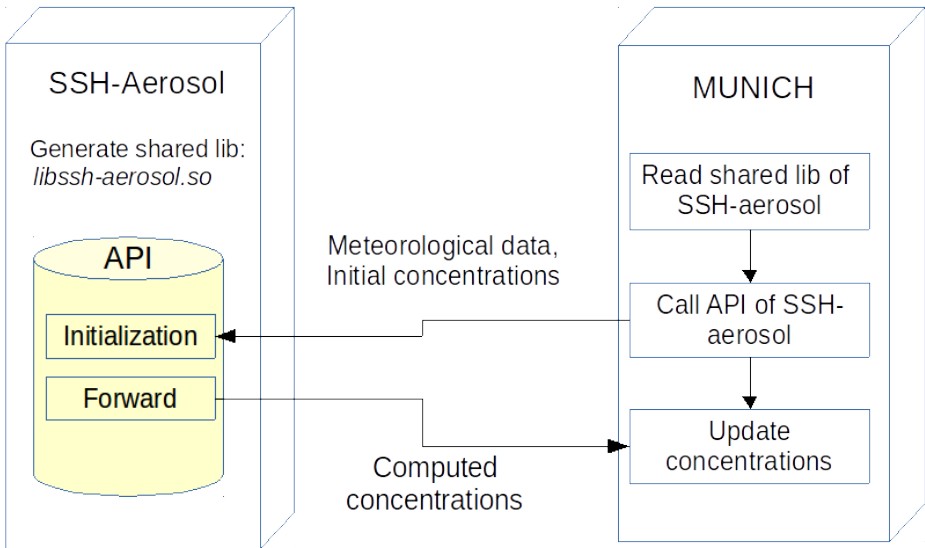

**Figure 3.** Schematic diagram of the coupling of SSH-aerosol with MUNICH using the API.

In MUNICH v2.0, the SSH-aerosol model (Sartelet et al., 2020) may be used to simulate both gas-phase chemistry and aerosol thermodynamics and dynamics (i.e., nucleation, condensation/evaporation, coagulation). SSH-aerosol is designed to be easily implemented into other models. It contains an Application Programming Interface (API), designed to allow for easy version updates. The API is used to implement SSH-aerosol v1.2 into MUNICH v2.0. The schematic diagram of coupling using the API is illustrated in Figure 3. A previous version of SSH-aerosol was implemented in MUNICH without using the API (Lugon et al., 2021a) and managed to represent well $PM_{10}$ and $PM_{2.5}$ concentrations in the streets of Paris, taking into account the formation of secondary inorganic and organic aerosols. The influence of secondary aerosol formation is presented in section 5.1.

## 2.5 Resuspension and deposition

Lugon et al. (2021b) introduced a new approach in MUNICH to estimate particle resuspension in streets. This approach strictly ensures the mass balance on the street surface. To do that, the accurate modelling of particle deposition and wash-off by water is mandatory. In MUNICH v2.0, the particle deposition is computed considering the available surface including pavement area and building walls as proposed in Cherin et al. (2015). For the particle wash-off, the amount of water on the street surface is computed from the meteorological conditions. Solubility of species is also an important factor for the wash-off parameterisation.

Modelling of the particle resuspension in MUNICH v2.0 requires an estimation of a resuspension factor. The resuspension factor is computed considering the traffic flow characteristics such as vehicle flow and speed, as detailed in Lugon et al. (2021b). The sensitivity of concentrations to deposition and resuspension is presented in section 6.





## 3 Reference test case

MUNICH v2.0 is applied to simulate the pollutant concentrations over a Paris suburb. The reference test case is set up over a district in the East part of Greater Paris between 22 March and 13 May 2014, which corresponds to a period where street measurements were performed (TRAFIPOLLU project, see the location of the station in Figure 4). The street network of the domain consists of 577 street segments and is displayed in Figure 4. The input data used for this study are now detailed. They are summarized in Table 1. The simulated concentrations are then compared to street observations.

### 3.1 Input data

#### 3.1.1 Traffic emissions

Traffic-related emissions in streets are computed using Pollemission (Sarica, 2021), which relies on emission factors from the COPERT methodology (COmputer Program to calculate Emissions from Road Transport, version 2019, EMEP/EEA, 2019) and the vehicle fleet. Emission factors are provided by the COPERT methodology for a wide range of vehicles types, depending

on fuel type and European emission standard. The COPERT methodology is used for emission factors from both exhaust and non-exhaust, i.e., wears of tyres and brakes and vehicle-induced abrasion of the road. Simulations using the dynamic traffic model SymuVia (Leclercq et al., 2007) provided, for each street segment of the network, the number of vehicles and speed profiles per hour and per category (passenger cars, light commercial vehicles, heavy-duty vehicles...) for a weekday and a weekend day. The vehicle fleet is mainly composed of passenger cars and light commercial vehicles, 77 % and 14 %

respectively on average. In each category, the breakdown by fuel and European standard is based on André et al. (2019). For each vehicle type, hourly profiles of vehicle flows and averaged speeds for a weekday and a weekend day are then used with COPERT emission factors to estimate the traffic emissions over the whole period (22 March to 13 May 2014).

Nitrogen oxides ($NO_x$) emission factors are speciated into NO and $NO_2$ using the fractions of $NO_2$ provided by the COPERT methodology for each vehicle type. Speciation of PM emission factors also follows the COPERT methodology by using the

fractions of black carbon (BC) and organic matter (OM) supplied for each vehicle type. The OM fraction of the PM emissions is assumed to be emitted as low-volatility organic compounds (LVOC) in the particle phase. If a fraction of PM remains after the BC and OM speciation, it is categorised as dust and unspeciated species. The PM size distribution at emission is assumed to be the same as in Lugon et al. (2021a, b), i.e. exhaust primary PM is assumed to be in the size bin [0.04 – 0.16 µm] while non-exhaust primary PM is coarser in the size bin [0.4 – 10 µm].

Non-methane volatile organic compounds (NMVOC) emission factors are computed as the difference between volatile organic compounds (VOC) and methane emission factors. In contrast to $NO_x$ and PM, the COPERT methodology presents five NMVOC speciation profiles given the fuel and category of the vehicle. These profiles include approximately 60 different species up to about C9 (9 carbon atoms) and lumped species for heavier compounds. Intermediate volatility organic compounds (IVOC) thus regroups the alkanes C10-C12, cycloalkanes, aromatics C9 and aromatics C10. Similarly, Semi volatility organic

compounds (SVOC) gathers the alkanes C>13 and aromatics C>13. Both IVOC and SVOC are emitted in the gas phase in the simulation, and the partitioning between the gas and particle phases is treated by the model when computing concentrations.



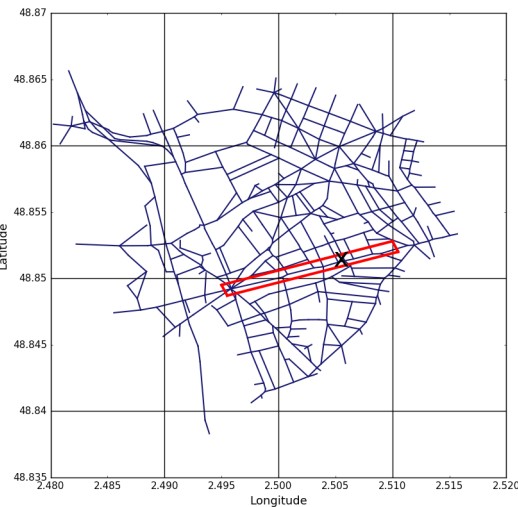

**Figure 4.** Street network of the domain. The street named "Boulevard Alsace Lorraine", where measurements were performed, is highlighted in the red box. The black cross mark corresponds to the location of the air monitoring station.

### 3.1.2 Geographic data

The widths of vehicle lane in the streets, street lengths and average building heights were obtained from the BD TOPO database ("Base de Données TOPOgraphiques", https://geoservices.ign.fr/bdtopo). The information for the sidewalk width and the high-

way shoulder width (the A86 highway passes through the modelling domain) is not available in the BD TOPO database. A width of 3 m is used for sidewalk of the streets, and 20 m (including two urban train lanes) for the shoulder of the A86 highway ) (Kim et al., 2018).

### 3.1.3 Regional-scale data

Meteorological data at a 1 km x 1 km horizontal resolution were obtained from Lugon et al. (2020), who conducted a simulation

using the Weather Research and Forecasting (WRF) model version 3.9.1.1 (Skamarock et al., 2008). In the WRF model setup, the single-layer urban canopy model (UCM) was used to represent the urban meteorological condition (Kusaka et al., 2001).

Background concentrations above the streets are obtained from the simulation results of the 3-dimensional chemistry-transport model Polair3D (Sartelet et al., 2007). The Polair3D simulation is presented in Sartelet et al. (2018); André et al. (2020). The same chemical scheme is used in the MUNICH simulation as in the Polair3D regional-scale simulation (CB05

with additional semi-volatile organic aerosols as detailed in Kim et al., 2011, Chrit et al., 2017 and Sartelet et al., 2020).





**Table 1.** Input data for MUNICH simulation.

| Data | Source | Reference |
| --- | --- | --- |
| Traffic emissions | Dynamic traffic model Symuvia and Pollemission for emission data | Leclercq et al. (2007), https://doi.org/10.5281/zenodo.5721253 |
| Geographic data | BD TOPO database ("Base de Données TOPOgraphique") | https://geoservices.ign.fr/bdtopo (in French) as used in Kim et al. (2018) |
| Meteorological data | WRF simulation (v3.9.1.1) | Lugon et al. (2020) |
| Background concentrations | Polair3d simulation | Sartelet et al. (2018); André et al. (2020) |

## 3.2 Simulated concentrations

The reference test case (Case-1) is performed for the period from 22 March to 13 May 2014 using the options of Table 2. In Figure 5, computed 24 h averaged concentrations are compared to the observed concentrations at the air monitoring station operated by Airparif during the TRAFIPOLLU project.

Two distinct statistical criteria are used to evaluate the model performance for hourly concentrations: an acceptance and a strict criteria (Hanna and Chang, 2012; Herring and Huq, 2018), see Table 3. The corresponding statistical indicators are defined in Appendix A1.

The hourly $NO_2$ concentrations estimate well the observations: the acceptance criteria are validated for all statistical indicators, and the strict criteria are validated for almost all indicators: the fractional bias (FB) is equal to -31%, while it should be

lower than 30% to satisfy the strict criteria. However, the NO concentrations are strongly underestimated, and do not satisfy the acceptance criteria. These discrepancies were observed in the previous studies (Kim et al., 2018; Lugon et al., 2020). The discrepancies in the simulation results using MUNICH v2.0 are reduced compared to those using MUNICH v1.0 but still high. The discrepancies can be explained by uncertainties in the traffic emission data, the vertical transfer at rooftop and the lifetime of NO (Kim et al., 2018; Lugon et al., 2020).

The statistical indicators for the simulated $PM_{10}$ and $PM_{2.5}$ concentrations are also satisfactory. For $PM_{10}$ concentrations both the acceptance and strict criteria are met for the different indicators. For $PM_{2.5}$ concentrations the acceptance criteria are met for the different indicators, but they do not meet the strict criteria for the FB, the normalized mean square error (NMSE) and the mean geometric bias (MG). The FB is equal to 34%, while it should be lower than 30% to satisfy the strict criteria. The overprediction for the $PM_{2.5}$ concentrations may be due to the uncertainties in the size distribution and non-exhaust emissions.

(Lugon et al., 2021a) showed that the observed and simulated $PM_{2.5}/PM_{10}$ ratios are lower at traffic stations (47% to 66%) than at urban background stations (67% to 76%) because of high non-exhaust emissions, mostly emitted as coarse particles. In the reference simulation, the observed $PM_{2.5}/PM_{10}$ ratio is 51%. However the simulated ratio is 89%.

Figure 6a shows the time-averaged concentrations over the simulation domain for the simulated $PM_{2.5}$ concentrations. The concentrations are high over the major streets where the emission rates are high.




**Table 2.** List of used options in the reference simulation.

| Option type | Used option |
|---|---|
| Solver | Explicit Trapezoidal Rule of order 2 (ETR) |
| Stationary approach | No |
| Turbulent vertical flux at the roof level | SCHULTE |
| Mean wind speed in the street canyon | Exponential |
| Wind speed at the roof level | SIRANE |
| Wind profile for deposition | MASSON |
| Resuspension | No |
| Chemistry (gas-phase chemistry and aerosol formation pathways) | Yes |
| Deposition | Yes |

**Table 3.** Statistical indicators of the comparison of simulated hourly concentrations to the measurements at the air monitoring station.

| | $NO_2$ | NO | $PM_{10}$ | $PM_{2.5}$ | Strict criteria | Acceptance criteria |
|---|---|---|---|---|---|---|
| Observation ($\mu g\,m^{-3}$) | 54.4 | 68.1 | 24.6 | 12.5 | | |
| Simulation ($\mu g\,m^{-3}$) | 39.8 | 22.9 | 19.7 | 17.6 | | |
| FB | -0.31 | -0.99 | -0.22 | 0.34 | -0.3 < FB < 0.3 | -0.67 < FB < 0.67 |
| NMSE | 0.28 | 1.54 | 0.21 | 0.44 | NMSE < 0.3 | NMSE < 0.6 |
| MFE | 0.46 | 1.00 | 0.38 | 0.47 | | |
| VG | 1.42 | 6.18 | 1.27 | 1.52 | VG < 1.6 | |
| MG | 0.68 | 0.30 | 0.80 | 1.38 | 0.7 < MG < 1.3 | |
| FAC2 | 0.75 | 0.20 | 0.84 | 0.74 | FAC2 $\geq$ 0.5 | FAC2 $\geq$ 0.3 |
| NAD | 0.22 | 0.49 | 0.19 | 0.25 | NAD < 0.3 | NAD < 0.5 |
| R | 0.58 | 0.76 | 0.68 | 0.48 | | |

Several simulations (sensitivity test cases) were performed to estimate the influence of the different model options on the computed concentrations. In each sensitivity test case, one parameterisation or process is modified with respect to the reference simulation. The characteristics of the simulations are listed in Table 4 and the available model options are explained in Appendix B. These sensitivity test cases are presented in the following sections. Domain-averaged normalized mean error (NME) between the sensitivity test case and the reference simulation is computed: a NME is computed for each street over the whole

simulation period and then averaged over the simulation domain. Larger domain-averaged NME means larger influence of the model option tested in the sensitivity test case.

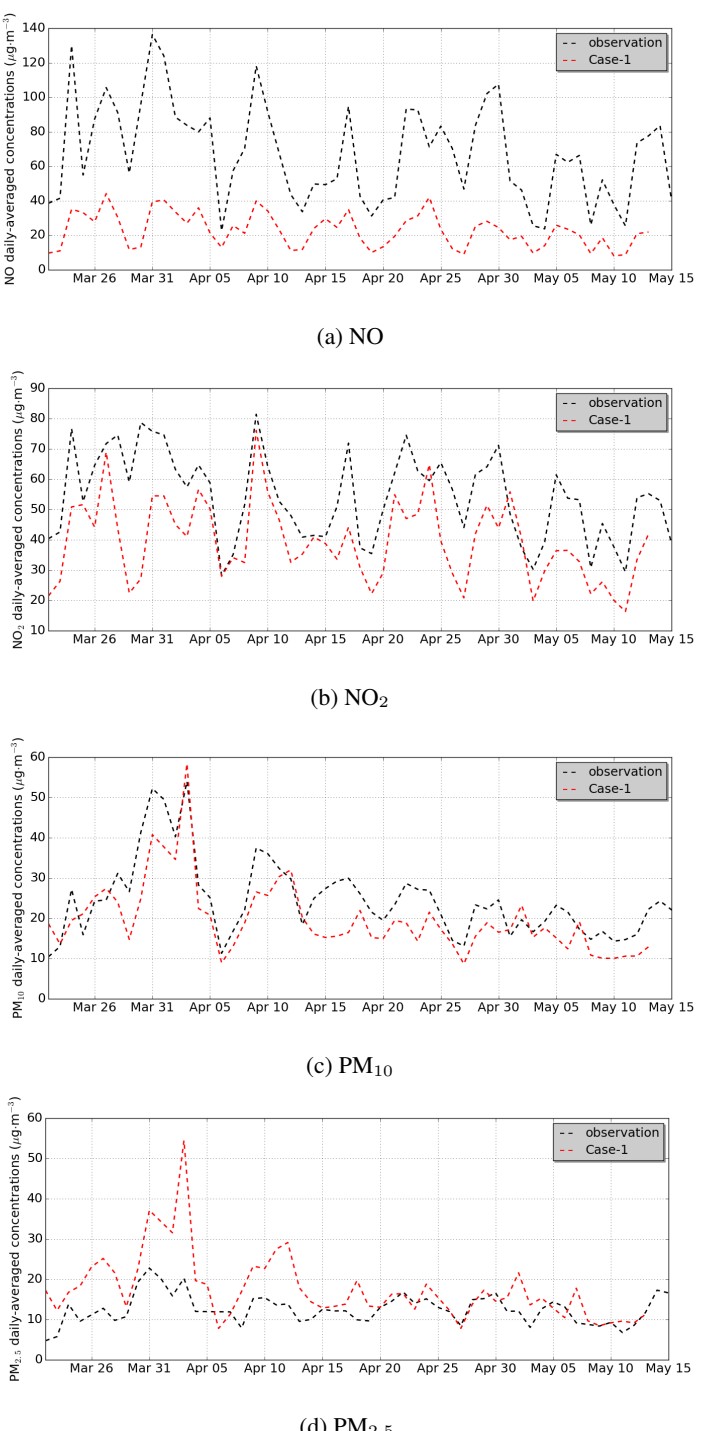

(a) NO

(b) NO$_2$

(c) PM$_{10}$

(d) PM$_{2.5}$

**Figure 5.** Comparison of daily concentrations (in $\mu g \, m^{-3}$) in Case-1 (reference) simulation to the measurements at the monitoring station for (a) NO and (b) NO$_2$ (c) PM$_{10}$, (d) PM$_{2.5}$.





(a) Case-1, PM$_{2.5}$ concentration

(b) NME between Case-10 and Case-1

(c) NME between Case-2 and Case-1

(d) NME between Case-11 and Case-1

**Figure 6.** PM$_{2.5}$ time-averaged concentrations (in $\mu g\,m^{-3}$) for the reference test case (Case-1, upper left panel) and temporal normalised mean error (NME, %) between sensitivity test cases and the reference test case, to quantify the average impact of parameterisations on the concentrations (Case-10 in the upper right panel, Case-2 in the lower left panel and Case-11 in the lower right panel).





**Table 4.** List of test cases and normalized mean error (NME, see Appendix A) between the sensitivity test case and the reference simulation (Case-1). The NME is computed for $PM_{2.5}$ and $NO_2$ for each street over the whole simulation period ahd then averaged over the whole simulation domain.

| Name of the test case | Changed option | NME ($PM_{2.5}$ / $NO_2$) |
|---|---|---|
| Case-1 (Reference) | - | - |
| Case-2 | Without chemistry | 13% / 11% |
| Case-3 | Without deposition | 1% / 2% |
| Case-4 | With resuspension | 1% / 0% |
| Case-5 | Stationary approach | 6% / 16% |
| Case-6 | Rosenbrock solver | 0% / 0% |
| Case-7 | Turbulent vertical transfer at the roof level: SIRANE | 1% / 4% |
| Case-8 | Mean wind speed in the street canyon: SIRANE | 1% / 1% |
| Case-9 | Turbulent mixing at intersection | 1% / 1% |
| Case-10 | Wind speed at the roof level: MACDONALD | 13% / 32% |

## 4 Influence of parameters related to transport

This section investigates the influence on concentrations of parameters related to transport, i.e. to the description of the wind velocity and the turbulence. Amongst the different parameterisation tested (Case-7, Case-8, Case-9 and Case-10), the estimation of the wind velocity at the roof level (Case-10) is the most influential. It directly impacts the strength of the wind speed within streets.

### 4.1 Wind velocity at the roof level

Two parameterisations may be used to compute the wind velocity at the roof level ($u_H$). In MUNICH v1.0, $u_H$ was computed with SIRANE parameterisation following Soulhac et al. (2011). The MACDONALD parameterisation is added in MUNICH v2.0, as detailed in section 2.2. The MACDONALD parameterisation is used in Case-10 simulation. An increase in both NO (NME of 55%) and $NO_2$ (NME of 32%) concentrations is observed with the MACDONALD parameterisation, see Figure 7.

For the comparison with the observation data, the MACDONALD parameterisation better simulates NO concentration than the SIRANE parameterisation. However the MACDONALD parameterisation overestimates the peaks of $NO_2$ concentrations.

Figure 6b shows the time-average NME (NME computed on the basis of temporal series) over the different street segments of the simulation domain for the $PM_{2.5}$ concentrations. The NME are high where the concentrations are high. The MACDONALD parameterisation leads to an increase of concentrations at the air monitoring station with a NME of about 15% and the maximum NME over the street domain is about 50% (13% over the whole domain).





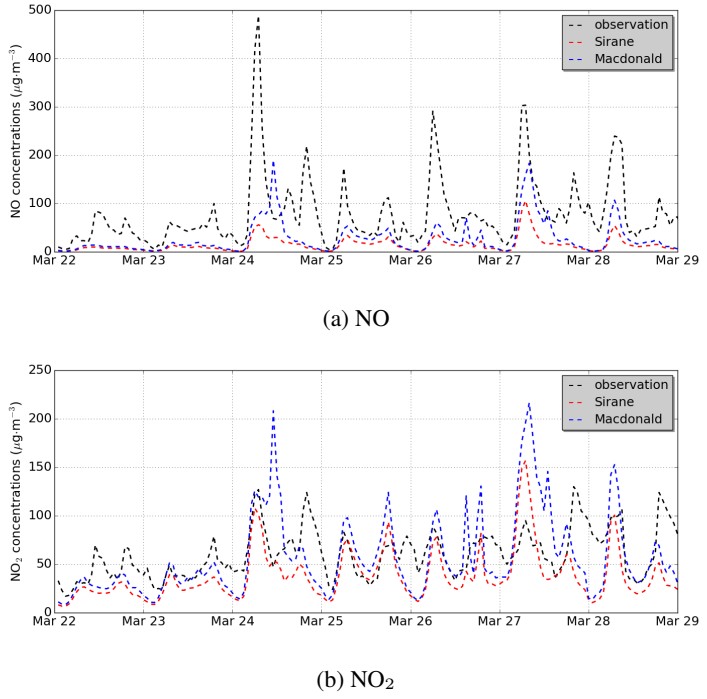

(a) NO

(b) NO$_2$

**Figure 7.** Comparison to observation of NO and NO$_2$ hourly concentrations (in $\mu g\,m^{-3}$) using two different parameterisations to compute the wind velocity at the roof level: SIRANE (Case-1) in red and MACDONALD (Case-10) in blue.

## 4.2 Turbulent transfer at the roof level

Two parameterisations are available to compute the turbulent vertical flux in MUNICH: SIRANE and SCHULTE. In the first one, the vertical flux is computed taking into account the street length and the street width. In the second one, the building height is also considered.

The sensitivity of the concentrations to this option is estimated by comparing the Case-7 simulation to the Case-1. The time-averaged NME, presented in Table 4, are low (1% for PM$_{2.5}$ and 4% for NO$_2$). However, the difference are more important for the peak concentrations with a maximum of 13% for PM$_{2.5}$ and 30% for NO$_2$ (see Figure 8).

Kim et al. (2018) showed that the vertical flux is higher with SCHULTE than SIRANE in areas with low buildings. On the contrary, the vertical flux is lower with SCHULTE than SIRANE in areas with tall buildings. The concentrations are then higher with the SIRANE parameterisation on the simulation domain where the building heights are low.

## 4.3 Wind speed formulation within the street and turbulent mixing at intersections

In the Case-8 simulation, the mean wind speed in the street canyon is calculated using the SIRANE parameterisation instead of the Exponential parameterisation in Case-1. Kim et al. (2018) showed that the impact of the mean speed using the SIRANE



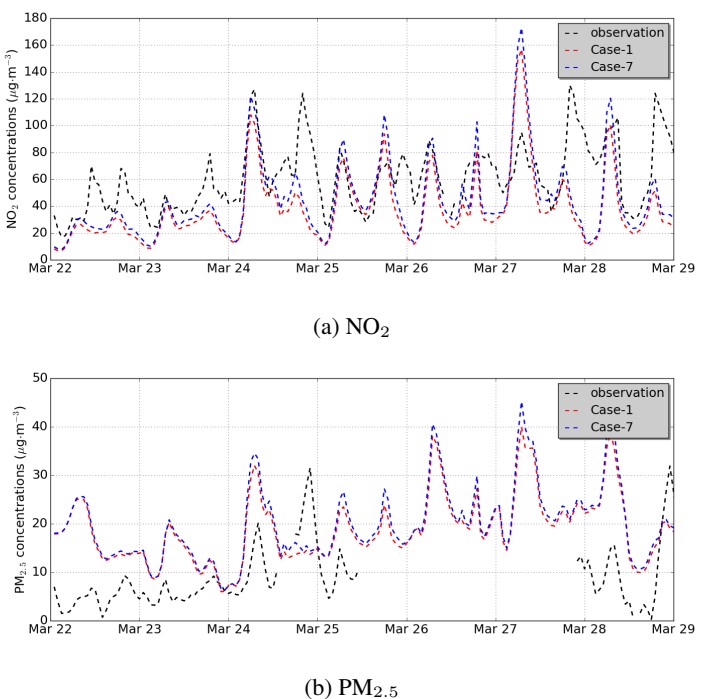

(a) NO$_2$

(b) PM$_{2.5}$

**Figure 8.** Comparison to observation of NO$_2$ and PM$_{2.5}$ hourly concentrations (in $\mu g\,m^{-3}$) using SCHULTE (Case-1) in red line and SIRANE (Case-7) in blue line.

or Exponential parameterisation is low for streets of low aspect ratio (about 1/3). The time-averaged NME between the Case-1 and Case-8 over the street network is also low: about 1% for PM$_{2.5}$ and 1% for NO$_2$.

Soulhac et al. (2011) suggested that the turbulent mixing at intersections can be represented by considering horizontal fluctuations in the wind direction. These horizontal fluctuations are parameterised using a Gaussian distribution of the wind direction, as detailed in Appendix B. The influence of the parameterisation of the turbulent mixing at intersections is tested in the Case-9 simulation. The time-averaged NME between Case-1 and Case-9 are low: about 1% for PM$_{2.5}$ and 1% for NO$_2$.

## 5 Influences of parameters related to secondary pollutant formation

### 5.1 Secondary gas and aerosols

In the simulation Case-2, the aerosol model SSH-aerosol is not used, and the pollutant concentrations are computed taking into account only emission, deposition and transport processes. Figure 6c shows the time-averaged NME over the simulation domain for the PM$_{2.5}$ concentrations between the Case-1 and Case-2 simulations. The NME over the whole domain for the PM$_{2.5}$ concentrations is 13%. Note that high NME are obtained over some major streets. Figure 9 presents the NME between the Case-1 and Case-2 simulations for the total PM$_{10}$ concentration and the concentrations of inorganic/organic aerosols. The



(a) PM$_{10}$

(b) Ammonium

(c) Nitrate

(d) Organic aerosols

**Figure 9.** Temporal NME (in %) between Case-1 and Case-2 for (a) PM$_{10}$, (b) Ammonium, (c) Nitrate and (d) Organic aerosols.

concentrations of PM$_{10}$ are reduced (NME of 11%) when chemistry and aerosol dynamics are not modelled. The reduction
280   is due to the absence of secondary inorganic and organic aerosol formation in the simulation Case-2. For inorganic aerosols,
the concentrations of ammonium and nitrate in Case-2 are reduced (NME of 24% and 5%, respectively). Very low change in
sulfate is obtained. For organic aerosols, the concentrations of particles that are formed from natural source are more reduced
(NME of 74%) than those formed from human activities (NME of 13%). The NME for total organic aerosol is 43%.

For the gas-phase species, the absence of conversion from NO to NO$_2$ by the chemical reactions in Case-2 leads to a
285   reduction of NO$_2$ in Case-2 (NME of 11%).

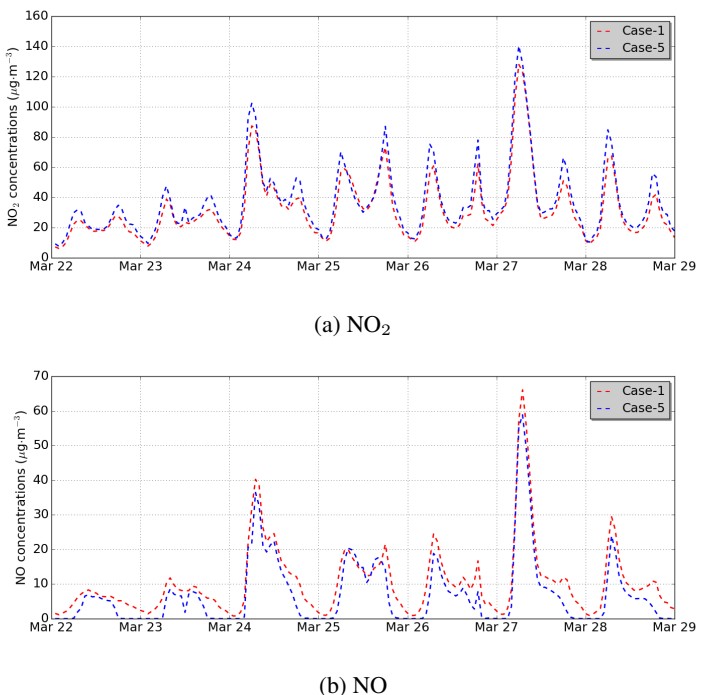

(a) NO$_2$

(b) NO

**Figure 10.** Comparison of (a) NO$_2$ and (b) NO hourly concentrations using the non-stationary approach (Case-1) in red and the stationary approach (Case-5) in blue. The concentrations are averaged over the whole simulation domain.

## 5.2 The non-stationary approach

In the simulation Case-5, the stationary hypothesis is assumed to compute the pollutant concentrations. As shown in Lugon et al. (2020) and Figure 10, higher concentrations of NO$_2$ are obtained in the simulation Case-5 than in Case-1 with a temporal NME of 35% for the Boulevard Alsace Lorraine and 16% in average over the domain). This increase in NO$_2$ concentration using the stationary hypothesis may be due to more conversion from NO to NO$_2$; the NO$_x$ concentration are similar between Case-1 and Case-5, the time-averaged NME are lower than 1%. Figure 11 presents the time-average NME in the concentrations simulated between Case-5 and Case-1. For PM$_{10}$ and PM$_{2.5}$, time-averaged NME are not as high as for NO$_2$. The NME is about 5% for both PM$_{10}$ and PM$_{2.5}$ in average over the domain.

For inorganic aerosols, ammonium concentrations using the stationary approach are larger (NME of 24%) than using the non-stationary approach. The concentration of nitrate in Case-5 is lower (NME of 13%). For organic aerosols, the differences are low (NME of 3%).

In Case-6, the Rosenbrock rather the ETR solver is used in the non-stationary approach. The simulated concentrations are not sensitive to the solver used (time-averaged NME lower than 1%).



(a) PM$_{10}$

(b) Ammonium

(c) Nitrate

(d) Organic aerosols

**Figure 11.** Temporal NME (in %) between Case-1 and Case-5 for (a) PM$_{10}$, (b) Ammonium, (c) Nitrate and (d) Organic aerosols.

## 6 Parameters related to deposition and resuspension

In the Case-3 simulation, deposition is not taken into account. Very low differences are obtained between the Case-1 and Case-3 simulations.

Particle dry-deposition has a negligible impact on PM concentration over the simulation domain (the time-averaged NME is 1% for PM$_{2.5}$, see Table 4. The gas-phase deposition parameterisation also has a low impact on sulfur dioxide (SO$_2$) and ozone (O$_3$) concentrations (the NME is about 1% in average). It is however important to notice this conclusion do not take into account the potential role of the urban vegetation in the deposition process (Janhäll, 2015). Moreover the average building height in the considered district is rather low. The deposition process could have a more significant impact for more densely built urban area.





In the Case-4 simulation, a parameterisation for particle resuspension is used. The amount of resuspended mass in MUNICH is limited by the deposited mass (Lugon et al., 2021b). Because the deposited mass is not significant in Case-3, the resuspended
mass in Case-4 is also low.

## 7   Sensitivity simulations

The street concentrations are strongly influenced by the building characteristics and by the traffic in the streets. To illustrate this influence, two sensitivity simulations are performed by arbitrarily modifying the building aspect ratio and by suppressing the traffic in a street.

**7.1   Influence of building aspect ratio**

The building aspect ratio, which is the ratio of building height to street width (H/W), is an important characteristic of streets, because it influences the turbulent transfer of pollutant at roof level and the vertical wind profile in the streets (Kim et al., 2018).

An additional sensitivity simulation (Case-11) is conducted to estimate the effect of the aspect ratio. The Case-1 reference
simulation is repeated by artificially modifying the building height and the street width. The street width is reduced by $\sqrt{3}$ and the building height is increased by $\sqrt{3}$ for all street segments. Therefore the aspect ratio is increased by 3. Modifying both the building height and the street width is important so that the volume of the street segments is not changed. Figure 6d presents that the NME between the Case-1 and Case-11 simulations are high where the $PM_{2.5}$ concentrations are high. Figure 12 shows the temporal variation of NO and $NO_2$ concentrations which are averaged on the whole simulation domain (Case-1 and Case-
11 simulations). The concentrations of NO and $NO_2$ in the Case-11 simulation are larger than those in the Case-1 simulation by the NME of 72% and 44%. The concentrations of $PM_{2.5}$ and $PM_10$ also increase in the Case-11 simulation by the NME of 16% and 17%, respectively. These larger concentrations are due to reduced turbulent transfer at roof level and reduced mean horizontal wind speed. Kim et al. (2018) estimated that the turbulent transfer decreases by 30% when the aspect ratio increases by 2.

**7.2   Effects of streets without cars**

Many European cities have taken Low-Emission Zone (LEZ) measures to reduce street-level air pollution. The effects of this type of measure can be simulated by reducing emissions in specific streets. An additional sensitivity simulation (Case-12) is conducted to estimate effects of emission reduction in a zone. In the Case-12 simulation, the setup of the reference simulation (Case-1) is used, but the emissions are set to zero in the Boulevard Alsace Lorraine (see Figure 4). It means all vehicles are
forbidden in this street.

Figure 13 shows the differences in $NO_2$ concentrations between Case-1 and Case-12 simulations. $NO_2$ concentration in Boulevard Alsace Lorraine with Case-12 simulation is lower than that with Case-1 simulation by 43% (Case-1: $44\,\mu g\,m^{-3}$ vs Case-12: $25\,\mu g\,m^{-3}$). It shows that pollutant concentrations are not negligible even though they are not emitted in the street.



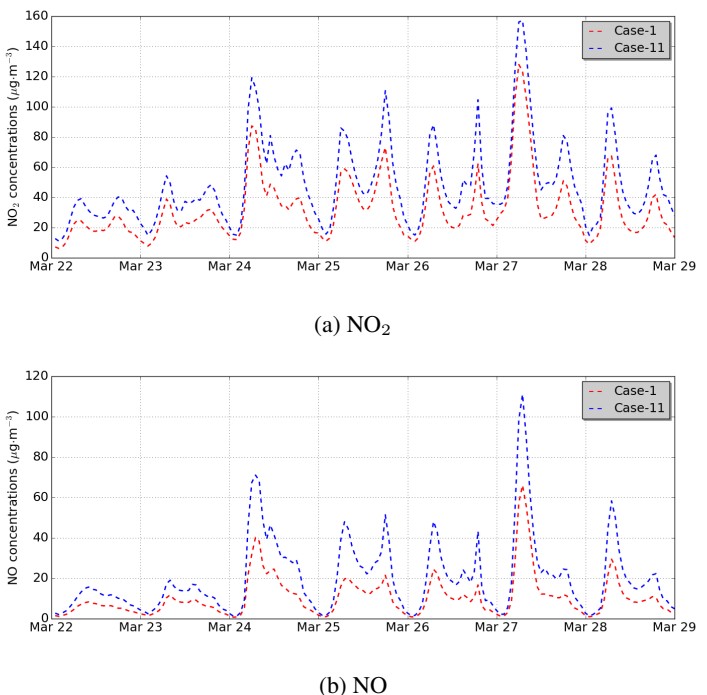

(a) NO$_2$

(b) NO

**Figure 12.** Comparison of the Case-1 hourly concentrations to the sensitivity simulation Case-11 modifying the building aspect ratio over the whole simulation domain: (a) NO$_2$ and (b) NO

It is due to the pollutant transfer from the overlying atmosphere and from the neighboring streets. However, they are strongly
reduced. For PM$_{10}$ and PM$_{2.5}$, the reduction is lower than those for NO$_2$ (18% for PM$_{10}$ and 16% for PM$_{2.5}$).

## 8 Conclusions

The street-network model MUNICH v2.0 is presented for multi-pollutant modelling in street canyons. A reference test case is set up in the East side of Greater Paris, where model to measurements are performed. NO$_2$, PM$_{2.5}$ and PM$_{10}$ are well modelled with MUNICH v2.0 compared to measurements.

A new parameterisation to compute the wind velocity at the roof level leads to an increase in PM$_{2.5}$ (13%) and NO$_2$ (32%) concentrations at the air monitoring station near traffic. The turbulent vertical transfer increases with the parameterisation taking into account the building height. It is due to low building heights in the street network studied here. This high sensitivity to wind velocity at the roof level underlines the importance of properly representing the transition from the regional to the street scale.

The SSH-aerosol model was implemented in MUNICH v2.0 for primary and secondary aerosol modelling in street canyons, taking into account gaseous chemistry leading to the formation of condensables, condensation/evaporation, nucleation and co-agulation. The PM$_{10}$ and PM$_{2.5}$ concentrations increased by 11% and 13%, respectively, if SSH-aerosol is used. This increase



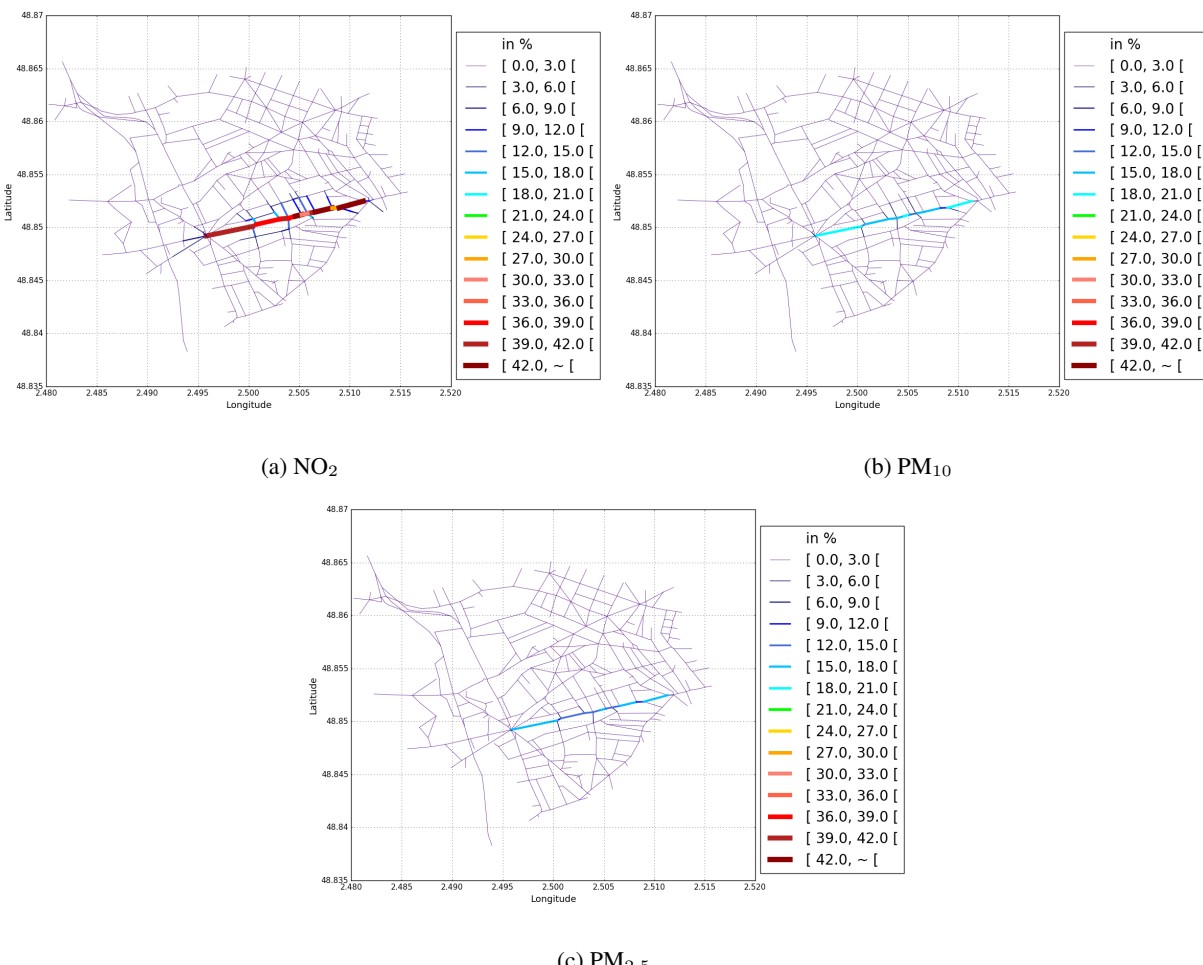

**Figure 13.** Temporal NME (in %) between Case-1 and Case-12 for (a) NO$_2$ (b) PM$_{10}$ and (c) PM$_{2.5}$.

is due to the formation of secondary inorganic and organic aerosols. The NO$_2$ concentration increased by 11% using SSH-aerosol. A non-stationary approach was developed to model reactive pollutants. In average over the street-network considered,

the non-stationary approach led to a decrease in NO$_2$ concentration by 16% compared to the stationary approach.

In comparison to MUNICH v1.0, parameterisations of particle deposition and resuspension were also added in MUNICH v2.0. However their impact on pollutant concentrations in the street canyons is low for the considered domain.

MUNICH may be easily be used with background concentrations from a regional air-quality model, in an one-way coupling approach. For the next step, the coupling between MUNICH v2.0 and regional air-quality model will be improved to consider

two-way coupling. The coupled model will be an updated version of Street-in-Grid model which computes both the pollutant concentrations within the street network and the average concentrations for the overlying atmosphere grid at same time.



*Code and data availability.* MUNICH v2.0 is available at Kim et al. (2022) or git repository at https://github.com/cerea-lab/munich. The configuration files and input data of the simulations and also scripts for figures are provided at https://doi.org/10.5281/zenodo.6167477. A user manual is available at http://cerea.enpc.fr/munich/doc/munich-guide-v2.pdf. The software requirements and the license information are

provided in the user manual.





# Appendix A: Statistical indicators

**Table A1.** Definitions of the statistical indicators.

| Indicators | Definitions |
| --- | --- |
| Fractional bias (FB) | $\dfrac{\overline{c}-\overline{o}}{(\overline{c}+\overline{o})/2}$ |
| Mean fractional bias (MFB) and mean fractional error (MFE) | $\dfrac{1}{n}\sum\limits_{i=1}^{n}\dfrac{c_i-o_i}{(c_i+o_i)/2}$ and $\dfrac{1}{n}\sum\limits_{i=1}^{n}\dfrac{\mid c_i-o_i\mid}{(c_i+o_i)/2}$ |
| Normalized mean square error (NMSE) | $\dfrac{\sum\limits_{i=1}^{n}(c_i-o_i)^2}{\sum\limits_{i=1}^{n}c_io_i}$ |
| Correlation coefficient (R) | $\dfrac{\sum\limits_{i=1}^{n}(c_i-\overline{c})(o_i-\overline{o})}{\sqrt{\sum\limits_{i=1}^{n}(c_i-\overline{c})^2}\sqrt{\sum\limits_{i=1}^{n}(o_i-\overline{o})^2}}$ |
| Geometrical mean squared variance (VG) | $\exp\left(\dfrac{\sum\limits_{i=1}^{n}\left(\left(\ln(c_i)-\ln(o_i)\right)^2\right)}{n}\right)$ |
| Mean geometric bias (MG) | $\exp\left(\dfrac{\sum\limits_{i=1}^{n}\left(\ln(c_i)-\ln(o_i)\right)}{n}\right)$ |
| Fraction of modeled values within a factor of two of observations (FAC2) | $0.5\leq c_i/o_i\leq 2$ |
| Normalized Absolute Difference (NAD) | $\dfrac{\frac{1}{n}\sum\limits_{i=1}^{n}\mid c_i-o_i\mid}{\overline{c}+\overline{o}}$ |
| Normalized Mean Error (NME) | $\dfrac{\frac{1}{n}\sum\limits_{i=1}^{n}\mid c_i-o_i\mid}{\overline{o}}$ |

$c_i$: modeled values, $o_i$: observed values, $n$: number of data.

$\overline{o}=\dfrac{1}{n}\sum\limits_{i=1}^{n}o_i$ and $\overline{c}=\dfrac{1}{n}\sum\limits_{i=1}^{n}c_i$





**Appendix B: Modelling options in MUNICH**

Several modelling options are available in MUNICH in order to handle the complexity and the computational time.

The options related with the modelling of the pollutant transport, deposition and resuspension are detailed here. Note that
the options linked to the modelling of chemical transformations and aerosol dynamics are presented in the article describing the SSH-aerosol model (Sartelet et al., 2020).

First of all, it is useful to present the main equations solved in MUNICH before the presentation of the options.

The time variation of the mass $M$ $\frac{dM}{dt}$ is computed using a transport-related term ($\frac{dM}{dt}|_{transp}$)and a chemistry-related term ($\frac{dM}{dt}|_{chem}$)

$$\frac{dM}{dt} = \frac{dM}{dt}|_{transp} + \frac{dM}{dt}|_{chem}.$$  (B1)

The transport-related term is computed as

$$\frac{dM}{dt}|_{transp} = Q_{inflow} + Q_{emis} - (Q_{outflow} + Q_{vert} + Q_{dep})$$  (B2)

where $Q_{inflow}$ is the incoming flux to the street, $Q_{emis}$ is the emission flux in the street, $Q_{outflow}$ is the outgoing flux from the street, $Q_{vert}$ is the vertical exchange flux at the roof level, $Q_{dep}$ is the deposition flux.

- `With_stationary_hypothesis`:

whether the stationary hypothesis is assumed or not (available options: yes or no)

If the stationary approach is used, the concentrations are computed in each street segment by assuming that $\frac{dM}{dt}|_{transp} = 0$. The non-stationary approach is recommended to model reactive species/pollutants. Note that the computation time increases by a factor 3 using the non-stationary approach for the reference test case (see section 2.3 and also 5.2).

- `Numerical_method_parameterisation`:

numerical solver (available options: ETR or Rosenbrock) The solver used to solve Eq. B2 with the non-stationary approach may either be the Explicit Trapezoidale Rule (ETR) or Rosenbrock. If the ETR solver is used, Eq. B2 is discretized as (Lugon et al., 2020):

$$C_s^{n+1} = C_s^n + \frac{\Delta t}{2}\left(F(C_s^n) + F(C_s^*)\right)$$  (B3)


$$C_s^* = C_s^n + \Delta t F(C_s^n)$$  (B4)

where $s$ represents a chemical species (gas or particle), $C_s^n$ is the concentration at time $t^n$, and $F(C_s^n)$ represents the time derivative of $C_s^n$ due to transport-related processes obtained by Eq. B2. The time step $\Delta t$ is adjusted by

$$\Delta t^{n+1} = \Delta t^n \sqrt{\frac{\Delta_0}{\Delta_1}}$$  (B5)



where $\Delta_1$ is the relative error and $\Delta_0$ is the relative error precision which is set to 0.01. The relative error $\Delta_1$ is computed as

$$\Delta_1 = \left\| \frac{C^{n+1} - C^*}{C^{n+1}} \right\|_2 \tag{B6}$$

where $C$ is the vector of concentration for all chemical species. The Euclidean norm is used to compute the relative error so that the error for all species is averaged.

The Rosenbrock solver is implemented to improve numerical stability of the non-stationary approach:

$$C_s^{n+1} = C_s^n + \frac{3}{2}\Delta t k_1 + \frac{1}{2}\Delta t k_2 \tag{B7}$$

$k_1$ and $k_2$ are computed as

$$(1 - \gamma \Delta t J)k_1 = F(C_s^n) \tag{B8}$$

$$(1 - \gamma \Delta t J)k_2 = F(C_s^{n+1} + \gamma k_1) - 2k_1 \tag{B9}$$

where $\gamma$ is $1 + \sqrt{2}/2$ and $J$ is a Jacobian matrix of Eq B2.

- `Transfer_parameterisation`:
  parameterisation to compute turbulent vertical mass transfer (available options: SIRANE or SCHULTE)

  The vertical flux, $Q_{\text{vert}}$ is formulated using the SIRANE option as follows:

$$Q_{\text{vert}} = \frac{\sigma_{\text{w}} W L}{\sqrt{2}\pi} \left( C_{\text{street}} - C_{\text{background}} \right) \tag{B10}$$

  where $C_{\text{background}}$ is the mean concentration above the street segment, $L$ is the street length, and $\sigma_{\text{w}}$ is the standard deviation of the vertical wind velocity at roof level, which depends on atmospheric stability.

  Using the SCHULTE option, the street aspect ratio ($a_r$, ratio of building height to street width) is taken into account:

  $$Q_{\text{vert}} = 0.45\sigma_{\text{w}} W L \left( \frac{1}{1 + a_r} \right) \left( C_{\text{street}} - C_{\text{background}} \right) \tag{B11}$$

where $a_r = H/W$

- `Building_height_wind_speed_parameterisation`:
  parameterisation to compute wind speed at the roof level (available options: SIRANE or MACDONALD)

  Using the SIRANE option (Soulhac et al., 2008), the wind speed at the roof level and at the center of the street ($u_M$) is computed as

$$u_M = u_* \sqrt{\frac{\pi}{\sqrt{2}\kappa^2 C} \left( Y_0(C) - \frac{J_0(C)Y_1(C)}{J_1(C)} \right)} \tag{B12}$$





where $u_*$ is the friction velocity, $J_0$, $J_1$ and $Y_1$ are Bessel functions. $\kappa$ is Von Karman constant. To compute the mean wind speed at the roof level over the street width ($u_H$), the horizontal wind speed variation of Soulhac et al. (2008) is considered. As discussed in Section 2.2, using the MACDONALD option, the wind speed at the roof level ($u_H$) is computed as

$$u_H = \frac{u_*}{\kappa} \ln\left(\frac{H - d_c}{z_{0_c}}\right) = u_{ref} \times \frac{\ln\left(\frac{H - d_c}{z_{0_c}}\right)}{\ln\left(\frac{z_{ref} - d_c}{z_{0_c}}\right)}. \tag{B13}$$

- Mean_wind_speed_parameterisation:

parameterisation to compute mean wind speed within the street canyon (available options: Exponential or SIRANE)

Using the Exponential option (Lemonsu et al., 2004), the wind speed within the street canyon is computed as

$$u_{street} = u_H |\cos(\varphi)| \frac{2}{a_r} \left(1 - \exp\left(\frac{a_r}{2}\left(\frac{z_0}{H} - 1\right)\right)\right) \tag{B14}$$

where $u_H$ represents the wind speed at the roof level and is computed by the option detailed above (see section 2.2), $\varphi$ is the angle between the street orientation and the wind direction, $z_0$ is the aerodynamic roughness of canyon surfaces.

Using the SIRANE option (Eq (1) in Soulhac et al. (2011)), the wind speed within the street canyon is computed as

$$u_{street} = u_M |\cos(\varphi)| \frac{\delta_i^2}{HW} \left(\frac{2\sqrt{2}}{C}(1-\beta)\left(1 - \frac{C^2}{3} + \frac{C^4}{45}\right) + \beta \frac{2\alpha - 3}{\alpha} + \left(\frac{W}{\delta_i} - 2\right)\frac{\alpha - 1}{\alpha}\right) \tag{B15}$$

where $\delta_i = min(H, W/2)$, $\alpha = ln\frac{\delta_i}{z_0}$, $\beta = exp\left(\frac{C}{\sqrt{2}}\left(1 - \frac{H}{\delta_i}\right)\right)$, $C$ is a solution of $\frac{z_0}{\delta_i} = \frac{2}{C} exp\left(\frac{\pi}{2}\frac{Y_1(C)}{J_1(C)} - 0.577\right)$

- With_horizontal_fluctuation:

whether turbulent mixing at intersection via the horizontal fluctuation of the wind direction is taken into account or not (available options: yes or no)

The horizontal fluctuation of the wind direction represents the turbulent mixing of the air across the intersection (Soulhac et al., 2008). The fluctuation is computed by the following steps:

1. When the fluctuation is not taken into account, the air flux from street i to street j, $P_{i,j}$ for the wind direction $\varphi$ is computed using the outgoing flux and the incoming flux at Eq. B2.

2. Compute N times, the air flux $P_{i,j}(\varphi + \sigma)$ for the wind direction $\varphi + \sigma$ where $\sigma$ is the fluctuation of the wind direction ranging from -20° to 20°. N is the number of $\sigma$ values. N is 10 when $\sigma$ is 20°

3. Compute the sum of the air flux.

$$P_{i,j} = \sum f(\sigma) P_{i,j}(\varphi + \sigma) \tag{B16}$$

where $f(\sigma)$ is a Gaussian distribution of the wind direction ranging from 0 to 1.

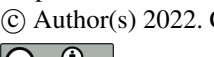



- `Deposition_wind_profile:`

wind profile option for dry deposition (available options: MASSON or MACDONALD)

The friction velocity is used to compute the deposition as follows:

$$u_* = \kappa \, exp \left( p(\frac{z}{h} - 1)/log(z/z_0) \right) \tag{B17}$$

where $p$ is the parameter for the wind profile.

The parameter $p$ may be computed using the MASSON option

$$p = 0.5 \, \frac{H}{W}, \tag{B18}$$

or using the MACDONALD option

$$p = 9.6 \, \lambda_f \tag{B19}$$

where $\lambda_f = \frac{H}{W + \frac{\lambda_p W}{1 - \lambda_p}}$ and $\lambda_p$ is the building density.

- `Particles_dry_velocity_option:`

parameterisation for aerosol deposition (available options: Zhang, Giardina, Venkatram or Muyshondt)

Using the Zhang option (Zhang et al., 2001), the deposition velocity is computed as

$$v_d = v_s + \frac{1}{R_{street}} \tag{B20}$$

where $R_{street}$ is the total resistance of the aerodynamic resistance of the street and the surface resistance and $v_s$ is the sedimentation velocity.

Using the Giardina option (Giardina and Buffa, 2018), the deposition velocity is computed as

$$v_d = v_s/(1 - exp^{v_s(Ra_{street} + \frac{1}{R_{eq}})}) \tag{B21}$$

where $Ra_{street}$ is the sum of the aerodynamic resistance of the street and $R_{eq}$ represents the resistance by the Brownian diffusion.

Using the Venkatram option (Venkatram and Pleim, 1999), the deposition velocity is computed as

$$v_d = v_s/(1 - exp^{v_s R_{street}}) \tag{B22}$$

Using the Muyshondt option (Muyshondt et al., 1996), the deposition velocity is computed as

$$v_d = v_s + v_{Re} \tag{B23}$$

where $v_{Re}$ represents the influence of the Reynolds number on the deposition velocity.



- `With_resuspension:`

  whether the resuspension is taken into account or not (available options: yes or no)

  Particle resuspension is computed based on a resuspension factor $f_{res}$

$$f_{res} = \sum_{v=1}^{2} N_v \left( \frac{u_v}{u_{ref}(r)} \right) f_{0,v} \tag{B24}$$

  where $v$ indicates the vehicle type, $N_v$ is the vehicle flow (vehicles per hour), $u_v$ is the vehicle speed ($km\ h^{-1}$), $u_{ref}(r)$ is the reference vehicle speed for the resuspension process ($km\ h^{-1}$), and $f_{0,v}$ the reference mass fraction of the resuspension process (per vehicle). It is detailed in Lugon et al. (2021b).

- `With_drainage_aerosol:`

  whether drainage is taken into account or not (available options: yes or no)

$$f_{wash} = \frac{1}{\delta t} \left( 1 - exp \left( -h_{drain,eff} \frac{g_{road} - g_{road,min}}{g_{road,min}} \right) \right) \tag{B25}$$

  where $\delta t$ is the time, $h_{drain,eff}$ is the drainage efficiency parameter, $g_{road}$ is the amount of water present on the street surface (mm), and $g_{road,min}$ is the minimum water content for the drainage process (mm).

  When this option is used, the wash-off factors are computed and associated with the precipitation. It is detailed in Lugon et al. (2021b).

- `With_chemistry:`

  whether chemistry is taken into account or not (available options: yes or no)

  SSH-aerosol model is used when this option is set to yes. The options for the chemistry model are defined in the namelist of SSH-aerosol, namelist.ssh.

*Author contributions.* YK, LL, KS, AM, YR, MV developed the software. MA provided the traffic data. YK conducted the simulations. YK and KS performed the analysis. YK and KS wrote the draft of the manuscript with contributions from AM and TS. All authors reviewed the final manuscript. YK, KS, YR and YZ were responsible for conceptualization, workshop and training of the software application.

*Competing interests.* The authors declare that they have no conflict of interest.

*Acknowledgements.* This work was partly funded by the Departement of Green Spaces and Environment (Mairie de Paris) and the École des Ponts ParisTech (grant CIFRE no. 2017/064), and by the sTREEt ANR project (ANR-19-CE22-0012).The authors acknowledge Airparif for providing the measured concentration data. YZ acknowledges funding from the NOAA Office of Climate AC4 Program (NA20OAR4310293).



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
