# Peer review of "MUNICH v2.0: A street-network model coupled with SSH-aerosol (v1.2) for multi-pollutant modelling"

_Geoscientific Model Development, 2022_

## Author Comment (AC1)

**Authors' reply to comments of anonymous reviewers on the manuscript "MUNICH v2.0: A street-network model coupled with SSH-aerosol (v1.2) for multi-pollutant modelling"**

Youngseob Kim[1], Lya Lugon[1], Alice Maison[1,2], Thibaud Sarica[1], Yelva Roustan[1], Myrto Valari[3], Yang Zhang[4], Michel André[5], and Karine Sartelet[1]

[1]CEREA, École des Ponts, EDF R&D, Marne-la-Vallée, France
[2]Université Paris-Saclay, INRAE, AgroParisTech, UMR EcoSys, Thiverval-Grignon, France
[3]Laboratoire de Météorologie Dynamique, Sorbonne Université, École Polytechnique, IPSL, École Normale Supérieure, CNRS, Paris, France
[4]Department of Civil and Environmental Engineering, Northeastern University, Boston, MA, USA
[5]Department COSYS, Université Gustave Eiffel, Bron, France

**Correspondence:** Youngseob Kim (youngseob.kim@enpc.fr), Karine Sartelet (karine.sartelet@enpc.fr)

We appreciate the reviewers for reading the manuscript attentively and giving helpful comments to improve it.

**1 Reply to anonymous reviewer #1's comments**

**1.1 General comments**

This paper presents the new version of the street-network model, Model of Urban Network of Intersecting Canyons and Highways version 2.0 (MUNICH v2.0) as well as its evaluation against observations (for a suburban area of Paris). This is very interesting state-of-the-art atmospheric chemistry transport model for the urban canopy. Its description and results of evaluation are well structured and thorough, which makes the paper relevant for the scope of the GMD journal.

However, the paper requires an extra work with respect to language. Plus, minor improvements and clarifications (specified below) are needed before publication.

**Our response**:

The English language has been revised and the manuscript has been clarified following the reviewer's comments.

**1.2 Specific comments**

1. Line 79: What does the sentence "An academic test case is set up in this section to illustrate how the pollutants are transported within the street network." exactly mean? What is the academic test?

    **Our response**:

    The text has been corrected in the revised manuscript.

"A demonstration test case is set up in section 2.1 to illustrate how the pollutants are transported within the street network from a single emission source. This test case is used for model validation from one version of MUNICH to the next."

2. Figure 1: Although you specified in the figure caption how you define the wind direction, it is still confusing when one look at the figure. Therefore, I recommend to add arrows indicating wind direction for every investigated wind direction (perhaps, a top row of 3 arrows or an arrow inside each subfigure (a, b, c, ...)).

**Our response**:

The arrows inside the figures are added. The text is corrected according to this change in Figure 1 as "Variation of pollutant concentrations in a street network depending on wind direction, which is indicated as dark-blue arrows."

3. Lines 17-18: Change s-1 to m×s-1

**Our response**:

The unit has been corrected.

4. Line 162: What is the actual location of observation site at "Boulevard Alsace Lorraine" (street or rooftop level)?

**Our response**:

The station is located on the ground, and the measurements were made with samples taken at a height of about 3 m. It is mentioned in the revised manuscript. "The reference test case is set up over a district in the eastern part of Greater Paris between 22 March and 13 May 2014, which corresponds to a period when street measurements were performed, with samples taken at a height of about 3 m."

5. Line 205: What is the offline coupling interval between Polair3D and WRF?

**Our response**:

The meteorological data from the WRF simulation are updated every hour in Polair3D simulation as MUNICH simulations. They are interpolated for the times between each hour.

The sentence

"The meteorological data from the WRF simulation are updated every hour in the MUNICH simulations."

has been corrected as follows:

"The meteorological data from the WRF simulation are updated every hour in the MUNICH simulations, and they are interpolated for the times between each hour."

6. Figure 4 or Figure 6: Perhaps, adding the mean wind direction (or better the wind rose) for the period of simulation (22 March and 13 May 2014) could be a good idea, since you have the test case (in Figure 1) indicating its influence for dispersion modelling in the urban canopy.

**Our response**:

A wind rose (Figure 1 below) and a related text have been added in the revised manuscript.

"Figure 4b shows the occurrence of wind direction over the simulation domain for the period from 22 March to 13 May 2014. The occurrence of wind direction is counted for wind that comes from each compass direction (N, NNE, NE, etc). South and Southwest winds are the prevailing winds during the simulation period."

[Figure]

**Figure 1.** Occurrence number of wind direction over the street network for the period from 22 March to 13 May 2014.

7. Figure 6 caption: What is the "temporal normalised mean error (NME)"? Is it time-averaged NME? If so, it seems not quite right definition/name as the temporal quantity implies time variability (like time-series).

    **Our response**:

    The word "temporal" was indeed inappropriate, as the NME is not a time-varying quantity. It has been removed, and the text has been corrected as follows:

    "PM$_{2.5}$ time-averaged concentrations (in $\mu g\,m^{-3}$) for the reference test case (Case-1, upper left panel) and normalised mean error (NME, %) between a sensitivity test case and the reference test case, which quantifies the average impact of parameterisations on the concentrations (Case-10 in the upper right panel, Case-2 in the lower left panel and Case-11 in the lower right panel)."

8. Lines 258-259: In the following sentence "However, the difference are more important for the peak concentrations with a maximum of 13% for PM2.5 and 30% for NO2 (see Figure 8)." do you mean the difference between observations and model output or between the Cases? The comparison of Cases does not exhibit so large differences (in particular for NO2).

**Our response**:

In this sentence, the results of the Case-1 and Case-7 simulations are compared. The difference of the time-averaged NME is low. However the peak values in the morning and evening rush hours show large differences.

The text has been clarified as follows:

"The sensitivity of the concentrations to this option is estimated by comparing the Case-7 simulation to the Case-1 simulation. Figure 8 presents a comparison to observation of $NO_2$ and $PM_{2.5}$ hourly concentrations in the Case-1 and Case-7 simulations. The time-averaged NME, between Case-1 and Case-7, presented in Table 4, are low (1% for $PM_{2.5}$ and 4% for $NO_2$). However, the differences between Case-1 and Case-7 are important for the peak concentrations during morning and evening rush hour. The peak concentrations of $NO_2$ in the Case-7 simulation (in blue line) are larger than those in the Case-1 simulation (in red line), by up to 30%. The largest differences are on March 24 evening and March 28 morning. For $PM_{2.5}$, the peak concentrations are less sensitive to the parameterisation of turbulent transfer, and the maximum difference between the two cases is 13% on March 27 morning."

9. Figures 6, 9, 11, 13: The NME metric gives only relative absolute change of a quantity. Thus, one cannot see the reduction or upswing in concentrations in the figures, which in turn means the statements in the text of the manuscript about the sign of changes (in sections 3-5) are not supported by the figures 6, 9, 11, 13.

   **Our response**:

   New figures (Figures 2 to 5 below) to show the absolute differences in the concentrations are added in the revised manuscript in Appendix C.

10. Line 326: Change $PM_10$ to $PM_{10}$

    **Our response**:

    It is corrected.

11. Lines 334-335: Since "vehicles are forbidden in this street", how is the traffic redistributed in the adjacent streets and what are the corresponding changes in traffic emissions (apart from "Boulevard Alsace Lorraine") in the Case-12? What would happen with the emissions in reality?

    **Our response**:

    In this illustrative test case, the background concentrations are the same as in the reference simulation. Therefore, the total emissions are the same in Case-12 and Case-1. We assume that the redistribution is done in nearby streets, but not adjacent.

    In the revised version, the sentences: "An additional sensitivity simulation (Case-12) is conducted to estimate the effects of emission reduction in a zone. In the Case-12 simulation, the setup of the reference simulation (Case-1) is used, but the emissions are set to zero in the Boulevard Alsace Lorraine (see Figure 4). It means all vehicles are forbidden in this

(a) Case-10 - Case-1

(b) Case-2 - Case-1

(c) Case-11 - Case-1

**Figure 2.** Differences in PM$_{2.5}$ time-averaged concentrations (in $\mu g\,m^{-3}$) between the reference test case (Case-1) and a sensitivity test case.

[Figure]

(a) PM$_{10}$

(b) Ammonium

(c) Nitrate

(d) Organic aerosols

**Figure 3.** Differences in time-averaged concentrations (in $\mu g \, m^{-3}$) between Case-2 and Case-1 (Case-2 - Case-1) for (a) PM$_{10}$ (b) Ammonium (c) Nitrate and (d) Organic aerosols.

[Figure]

**Figure 4.** Differences in time-averaged concentrations (in $\mu g\,m^{-3}$) between Case-5 and Case-1 (Case-5 - Case-1) for (a) $PM_{10}$ (b) Ammonium (c) Nitrate and (d) Organic aerosols.

[Figure]

(a) NO$_2$

(b) PM$_{10}$

(c) PM$_{2.5}$

**Figure 5.** Differences in time-averaged concentrations (in $\mu g\,m^{-3}$) between Case-12 and Case-1 (Case-12 - Case-1) for (a) NO$_2$ (b) PM$_{10}$ (c) PM$_{2.5}$.

95         street." are replaced by "An additional sensitivity simulation (Case-12) is conducted to estimate the effects of emission reduction in a street. In the Case-12 simulation, the setup of the reference simulation (Case-1) is used, but the emissions are set to zero in the Boulevard Alsace Lorraine (see Figure 4). It means all vehicles are forbidden in this street. The background concentrations are the same in Case-12 as in the reference simulation, meaning that the total emissions are the same in both simulations. However, we assume that traffic is redistributed in nearby streets of Boulevard Alsace

100         Lorraine, but not directly adjacent."

12. Lines 48-49: The following statement is vague: "the importance of properly representing the transition from the regional to the street scale". What do you mean by the "property"?

**Our response**:

The word "properly" has been replaced by "accurately", and the sentence "This high sensitivity to wind velocity at the
105         roof level underlines the importance of properly representing the transition from the regional to the street scale." has been replaced by "This high sensitivity to wind velocity at the roof level underlines the importance of meteorological down-scaling to accurately represent the transition from the regional to the street scale."

**2    Reply to anonymous reviewer #2's comments**

**2.1    General comments**

110 The authors present the second version of the air quality street model titled MUNICH. In this new version, the original model has been improved in terms of its numerical solution for atmospheric chemistry (the steady-state assumption has been removed), its atmospheric dynamics parametrizations, and its coupling with a state-of-the-science organic aerosol model. These improvements are significant as shown by the simulation results presented in the paper. Although the coupling of models (such as MUNICH and SSH here) seems conceptually straightforward, the actual development of an integrated model requires care
115 and effort to ensure internal consistency within the new model. Here, it appears that the authors have been meticulous in their model development work, as exemplified by the many comparisons among the various options now available in the model for the treatment of atmospheric chemistry, aerosol processes, and atmospheric dynamics. Therefore, I recommend publication with minor corrections needed to address the following comments.

     As currently written, the paper lacks scientific insights regarding the results of the various simulations being reported.
120 Although the scientific aspects of the modeling results have been presented in earlier publications by the authors, this paper should be a stand-alone document and summaries of key scientific results should be provided here. For example, on line 340, the authors state that PM concentrations decrease less than those of NO2. This is a fact, but the reasons leading to this result should be provided. This result is due to differences in the atmospheric processes leading to PM and NO2 concentrations (longer atmospheric lifetimes and, therefore, larger contributions of the background in the case of PM leading to lower contributions
125 of local PM emissions). Those differences should be mentioned explicitly in order to explain the modeling results.

Furthermore, I assume that potential users would want some guidance on which model options are the most appropriate depending on the application considered. Comparisons of modeling results with observations are very useful and definitely provide confidence in the model performance. However, they cannot be used solely to discriminate among various model options, because better agreement with measurements may result from compensation of errors (e.g., uncertainties in model inputs such as emissions and meteorology). Nevertheless, the authors must have some clear ideas on which options are considered best. (Indeed, they make such a recommendation regarding the use of the steady-state approach; however, that recommendation appears only in an appendix.)

Such recommendations may be based on modeling results (for example, they recommend not to use the steady-state assumption for chemically-reactive pollutants), theoretical considerations (an algorithm may be more comprehensive than another), comparisons with more advanced models (for example, with a CFD model in the case of some atmospheric dynamics parametrizations), etc. If the authors cannot decide whether one option is better than another, they could simply say so. The results of such a discussion would be very useful to potential users and could help avoid potential misuse of the model. Recommendations could be presented in each relevant section and then summarized in the conclusion. In addition, the modeling options should be summarized in a table, with brief descriptions of their pros and cons, along with recommendations for those to be used in a base case simulation (reference to Appendix B for more details could be included in that table). Furthermore, some standard model configurations could be provided, for example, a configuration to emulate the original SIRANE model, a configuration to simulate chemically-inert pollutants, another for chemically-reactive gaseous pollutants and another for all (gaseous and particulate) pollutants, etc.

**Our response**:

The scientific insights mentioned by the reviewer were added in the revised manuscript: "This higher contribution of street emissions to concentrations for $NO_2$ than for PM is due to differences in the atmospheric processes leading to PM and $NO_2$ concentrations (longer atmospheric lifetimes and, therefore, larger contributions of the background leading to lower contributions of local PM emissions)."

Recommendations about the modelling options have been added in section 2, at the end of each paragraph describing the modelling options.

**2.2  Specific comments**

Specific comments follow, including comments related to the modeling options.

1. Abstract, lines 9-10: This sentence should reflect the fact that deposition on vegetation (e.g., trees) is not considered in this study. For example: "deposition on built surfaces..."

   **Our response**:

   The sentence "The impact of particle deposition and resuspension on pollutant concentrations in the street canyons is low." has been replaced by "The impact of particle deposition on build surfaces and road resuspension on pollutant concentrations in the street canyons is low."

2. Line 14: Street-canyons are not limited to European cities. In other words, the authors should not restrict the use of their model to European countries; as a matter of fact, applications of MUNICH to cities in South America and Asia have been published in the scientific literature.

   **Our response**:

   The word "European" has been removed.

3. In Section 2, the authors present some improvements to the model and one assumes that the latter option is recommended, i.e., the MacDonald algorithm for the wind speed at roof level and the dynamic solution for the chemistry/transport equations. It seems that the older options (Sirane algorithm for the wind speed and steady-state solution) are still available in MUNICH 2.0 and the authors should explain under which circumstances they recommend using them.

   **Our response**:

   Two parameterisations for the computation of the roof-level wind speed are available in MUNICH: Sirane (Soulhac et al., 2011) or Macdonald (Macdonald et al., 1998). The Macdonald parameterisation was added recently because we observed that Sirane overestimates the roof-level wind speed in comparison to the CFD simulation results of Maison et al. (2022). Therefore, we would recommend using the Macdonald parameterisation. However, by error compensation, simulations with Sirane could give better scores compared to observations for some applications. Therefore, the parameterisation is kept in MUNICH.

4. In Section 4, various options for atmospheric dynamics are investigated. The roof-level wind speed algorithms, which were already mentioned in Section 2, are compared. Which one is recommended?

   **Our response**: As detailed above, we recommend to use the Macdonald parameterisation. The following sentences have been added to the revised version, at the end of section 4.1: "Because the MACDONALD parameterisation better estimated the roof-level wind speeds than the SIRANE one, in comparison to the CFD simulation results of Maison et al. (2022), the MACDONALD parameterisation is recommended in MUNICH. However, because of uncertainties on the regional wind speed and friction velocity, simulations with the SIRANE parameterisation could give better scores compared to observations for some applications. "

5. Two algorithms are available for the calculation of turbulent vertical mass transfer at roof-level, the original Sirane parametrization and that of Schulte et al. The latter includes more information regarding the street configuration; do the authors recommend it?

   **Our response**: Indeed, the Schulte et al. parameterisation (Schulte et al., 2015) for the turbulent vertical mass transfer at roof-level includes an additional dependence to the street aspect ratio compared to Sirane (Soulhac et al., 2011). Based on the comparison to CFD simulations Maison et al. (2022), we recommend using the Schulte et al. parameterisation for the turbulent vertical mass transfer at roof-level.

The following sentences have been added at the end of section 4.2: "Because the SCHULTE parameterisation for the turbulent vertical mass transfer at roof-level includes an additional dependence to the street aspect ratio compared to SIRANE one, leading to better comparisons to the CFD simulations of Maison et al. (2022), the SCHULTE parameterisation is recommended in MUNICH."

6. The wind speed within the street may be calculated according to various algorithms; a previous paper by the authors included comparisons of MUNICH with a computational fluid dynamics (CFD) model and the results of that previous work could be mentioned at this point and possibly be used as a basis for some recommendations.

   **Our response**: The comparison of MUNICH transport parameterisation to CFD simulations shows that the exponential profile overestimates the wind speed in the street especially at the bottom of the street because the no-slip condition on the ground is not satisfied Maison et al. (2022). Therefore, we would recommend to use the SIRANE parameterisation for the horizontal wind speed within the street.

   The following sentences have been added at the end of section 4.3: "Because the comparison to CFD simulations shows that the exponential profile overestimates the wind speed in the street especially at the bottom of the street (Maison et al., 2022), the SIRANE parameterisation is recommended for the horizontal wind speed within the street. Taking into account horizontal fluctuations in the wind direction is not necessary, because of its low influence on concentrations."

7. In Section 5, the authors investigate the effect of chemical transformations (including aerosol processes) on air pollutant concentrations in the streets. The use or not of chemical transformations is investigated for PM concentrations and NO2. As expected, gas-phase chemistry must be taken into account for the conversion of primary NO to secondary NO2 by ozone titration; this should be explicitly stated (currently, the result is mentioned, but without any specific recommendation on whether chemistry should be included or not).

   **Our response**:

   The following sentences have been added at the end of section 5.1: "As a large fraction of $NO_2$ is secondary, formed from the conversion of primary NO by ozone titration (Lugon et al., 2020), it is crucial to take gas-phase chemistry into account to accurately represent $NO_2$ concentrations."

8. The formation of secondary PM is significant, especially for organic aerosols. This result may seem counterintuitive at first, since one would expect oxidant levels to be particularly low in street canyons (due to titration of ozone by NO, see above) and, therefore, oxidation of NO2 to nitrate, SO2 to sulfate, and VOC to SOA to be slow. The authors have investigated this issue in a previous publication and they should mention the causes for secondary PM formation obtained here in a street-canyon scenario (e.g., reaction of NH3 emitted from vehicular traffic with existing HNO3). One assumes that it is preferable to include chemical transformations to obtain a better simulation of PM concentrations, but the authors should state it explicitly.

   **Our response**:

The following sentences have been added at the end of section 5.1: "The inorganic and organic concentrations of PM are strongly influenced by aerosol dynamics, mostly because of the condensation/evaporation process (e.g. $NH_3$ from traffic emission condenses with existing $HNO_3$). However, the coagulation process also needs to be taken into account to accurately represent the particle size distribution (Lugon et al., 2021a)."

The following sentences have been added at the end of section 5.2: "For secondary compounds, such as $NO_2$, inorganic and organic aerosols, it is crucial to use the non-stationary approach, as it ensures numerical stability and strongly affects the concentrations."

9. Regarding atmospheric deposition, it is mentioned that deposition on vegetation could be an important process. Are there plans to include this process in a future version of the model? There are several options available in MUNICH to simulate dry deposition (Appendix B): Zhang et al., Venkatram and Pleim, Giardina and Buffa, Muyshondt et al. What are the pros and cons of those various algorithms and do the authors recommend one in particular?

**Our response**: Recent developments aimed to parameterise the aerodynamic effect of tree crown (https://acp.copernicus.org/preprints/acp-2022-287/acp-2022-287.pdf), and the dry deposition of gaseous pollutants and aerosols on tree leaves (based on Zhang et al. (2001, 2002, 2003) and Giardina and Buffa (2018)) in MUNICH. They are not integrated in MUNICH v2.0, but they will be in the next MUNICH version.

We recommend the Venkatram option to calculate particle dry deposition, as it was used by Lugon *et al.,* (2021). The Venkatram option performs better than the other options to calculate black carbon dry deposition, with a good correlation between measured and simulated particle deposition over the street surface.

The following text has been added in Appendix. "Lugon et al. (2021b) shows that the Venkatram option performs better than the other options to calculate black carbon dry deposition, with a good correlation between measured and simulated particle deposition over the street surface."

10. Regarding resuspension and removal of deposited PM by rain, one may assume that, based on their earlier work, the authors recommend including those processes when simulating PM. This could be stated explicitly; then, the options of not including those processes are available to investigate their importance on PM concentrations in street canyons. They can of course be ignored if only gases are simulated.

**Our response**:

The following sentences have been added at the end of section 6: "Dry-deposition on urban surfaces and resuspension have a low impact on concentrations in Paris. However, wet-deposition by rain may have a large impact during rainy days and should be considered (Roustan et al., 2010; Vivanco et al., 2018)."

11. Although this paper is rather well organized and easy to read, the authors must carefully go through the text to correct grammatical and vocabulary errors before final submittal.

**Our response**:

The English language has been revised.

**References**

Giardina, M. and Buffa, P.: A new approach for modeling dry deposition velocity of particles, Atmos. Environ., 180, 11–22, https://doi.org/10.1016/j.atmosenv.2018.02.038, 2018.

Lugon, L., Sartelet, K., Kim, Y., Vigneron, J., and Chrétien, O.: Nonstationary modeling of $NO_2$, NO and $NO_x$ in Paris using the Street-in-Grid model: coupling local and regional scales with a two-way dynamic approach, Atmos. Chem. Phys., 20, 7717–7740, https://doi.org/10.5194/acp-20-7717-2020, 2020.

Lugon, L., Sartelet, K., Kim, Y., Vigneron, J., and Chrétien, O.: Simulation of primary and secondary particles in the streets of Paris using MUNICH, Faraday Discuss., 226, 432–456, https://doi.org/10.1039/D0FD00092B, 2021a.

Lugon, L., Vigneron, J., Debert, C., Chrétien, O., and Sartelet, K.: Black carbon modeling in urban areas: investigating the influence of resuspension and non-exhaust emissions in streets using the Street-in-Grid model for inert particles (SinG-inert), Geosci. Model Dev., 14, 7001–7019, https://doi.org/10.5194/gmd-14-7001-2021, 2021b.

Macdonald, R., Griffiths, R., and Hall, D.: An improved method for the estimation of surface roughness of obstacle arrays, Atmos. Environ., 32, 1857–1864, https://doi.org/10.1016/S1352-2310(97)00403-2, 1998.

Maison, A., Flageul, C., Carissimo, B., Tuzet, A., and Sartelet, K.: Parametrization of Horizontal and Vertical Transfers for the Street-Network Model MUNICH Using the CFD Model Code_Saturne, Atmosphere, 13, 527, https://doi.org/10.3390/atmos13040527, 2022.

Roustan, Y., Sartelet, K., Tombette, M., Debry, É., and Sportisse, B.: Simulation of aerosols and gas-phase species over Europe with the Polyphemus system. Part II: Model sensitivity analysis for 2001, Atmos. Environ., 44, 4219–4229, https://doi.org/10.1016/j.atmosenv.2010.07.005, 2010.

Schulte, N., Tan, S., and Venkatram, A.: The ratio of effective building height to street width governs dispersion of local vehicle emissions, Atmos. Environ., 112, 54 – 63, https://doi.org/10.1016/j.atmosenv.2015.03.061, 2015.

Soulhac, L., Salizzoni, P., Cierco, F.-X., and Perkins, R.: The model SIRANE for atmospheric urban pollutant dispersion; part I, presentation of the model, Atmos. Environ., 45, 7379 – 7395, https://doi.org/10.1016/j.atmosenv.2011.07.008, 2011.

Vivanco, M. G., Theobald, M. R., García-Gómez, H., Garrido, J. L., Prank, M., Aas, W., Adani, M., Alyuz, U., Andersson, C., Bellasio, R., Bessagnet, B., Bianconi, R., Bieser, J., Brandt, J., Briganti, G., Cappelletti, A., Curci, G., Christensen, J. H., Colette, A., Couvidat, F., Cuvelier, C., D'Isidoro, M., Flemming, J., Fraser, A., Geels, C., Hansen, K. M., Hogrefe, C., Im, U., Jorba, O., Kitwiroon, N., Manders, A., Mircea, M., Otero, N., Pay, M.-T., Pozzoli, L., Solazzo, E., Tsyro, S., Unal, A., Wind, P., and Galmarini, S.: Modeled deposition of nitrogen and sulfur in Europe estimated by 14 air quality model systems: evaluation, effects of changes in emissions and implications for habitat protection, Atmos. Chem. Phys., 18, 10 199–10 218, https://doi.org/10.5194/acp-18-10199-2018, 2018.

Zhang, L., Gong, S., Padro, J., and Barrie, L.: A size-segregated particle dry deposition scheme for an atmospheric aerosol module, Atmos. Environ., 35, 549–560, https://doi.org/10.1016/S1352-2310(00)00326-5, 2001.

Zhang, L., Moran, M. D., Makar, P. A., Brook, J. R., and Gong, S.: Modelling gaseous dry deposition in AURAMS: a unified regional air-quality modelling system, Atmos. Environ., 36, 537–560, https://doi.org/10.1016/S1352-2310(01)00447-2, 2002.

Zhang, L., Brook, J. R., and Vet, R.: A revised parameterization for gaseous dry deposition in air-quality models, Atmos. Chem. Phys., p. 16, https://doi.org/10.5194/acp-3-2067-2003, 2003.

---

## Author Response (AR2)

**Authors' reply to comments of the editor on the manuscript "MUNICH v2.0: A street-network model coupled with SSH-aerosol (v1.2) for multi-pollutant modelling"**

Youngseob Kim[1], Lya Lugon[1], Alice Maison[1,2], Thibaud Sarica[1], Yelva Roustan[1], Myrto Valari[3], Yang Zhang[4], Michel André[5], and Karine Sartelet[1]

[1]CEREA, École des Ponts, EDF R&D, Marne-la-Vallée, France
[2]Université Paris-Saclay, INRAE, AgroParisTech, UMR EcoSys, Thiverval-Grignon, France
[3]Laboratoire de Météorologie Dynamique, Sorbonne Université, École Polytechnique, IPSL, École Normale Supérieure, CNRS, Paris, France
[4]Department of Civil and Environmental Engineering, Northeastern University, Boston, MA, USA
[5]Department COSYS, Université Gustave Eiffel, Bron, France

**Correspondence:** Youngseob Kim (youngseob.kim@enpc.fr), Karine Sartelet (karine.sartelet@enpc.fr)

We appreciate the reviewers for reading the manuscript attentively and giving helpful comments to improve it.

**1 Reply to the editor's comments**

1. The reviewer 2 pointed out that the paper lacks scientific insights regarding the results of the various simulations being reported and presented PM/NO2 example. You succesfully added more description on PM/NO2 part but not elsewhere. Thus you should carefully revise the manuscript and add addition description to those parts where it is missing. This applies at least to sections 4.1, 4,2 and 5.1.

   **Our response**:

   The manuscript has been revised to add the following texts in Section 4.1

   "This increase is due to lower wind velocity at the roof level with the MACDONALD parameterisation, which leads to a lower dispersion of $NO_x$ from the streets where the air monitoring station is located."

   in Section 4.2

   "The computation of the vertical flux depends on the gradient between the street concentration and the background concentration in both parameterisations. The gradient is large during the rush hours because of high traffic emissions. This large gradient leads to a large difference in the vertical flux between Case-1 and Case-7 during the rush hours."

   "$PM_{2.5}$ concentrations are less sensitive to most parameterisations than $NO_2$ concentration in our simulations except for the Case-2 simulation (see Table 4). This is due to a larger contribution of background emissions for $PM_{2.5}$ than $NO_2$."

   and in Section 5.1

"Lugon et al. (2021a) showed that the average impacts of secondary aerosol formation on $PM_{2.5}$ concentrations over the streets in Paris are 12% for organic aerosol and 7% for inorganic aerosol."

"Very low change in sulfate is obtained because the sulfate in the streets in mainly imported from the background (Lugon et al., 2021a)."

"It is however worth noting that the emission from the urban vegetation is not taken into account in this result."

2. L268: MACDONAL is missing D.

**Our response**:

It has been corrected in the revised manuscript.

3. To clarify reading of the figures, I recommend you add panel information (a-..) to the figure texts where it is missing. This applies at least to Figure 1 (instead of upper and lower three cases use a-c, d-e), Figure 2 ( plan and frontal, I would also add abbreviations of them to to the text as these are not necessarily familiar for all readers), Figure 7 and Figure 8.

**Our response**:

As the Editor's recommendation, the figure texts have been corrected as follows:

in Figure 1,

"Variation of pollutant concentrations in a street network depending on wind direction, which are indicated as arrows in dark blue. The wind speed is $5\,\mathrm{m\,s^{-1}}$ for (a), (b) and (c) and $10\,\mathrm{m\,s^{-1}}$ for (d), (e) and (f)."

in Figure 2,

"(a) $d_c/\overline{H}$ and (b) $z_{0_c}/\overline{H}$ as a function of the plan and frontal area densities ($\lambda_P$ and $\lambda_F$) calculated by Eq. 3"

in Figure 7,

"Comparison to observation of (a) NO and (b) $NO_2$ hourly concentrations (in $\mu\mathrm{g\,m^{-3}}$) using two different parameterisations to compute the wind velocity at the roof level: SIRANE (Case-1) in red and MACDONALD (Case-10) in blue."

in Figure 8,

"Comparison to observation of (a) $NO_2$ and (b) $PM_{2.5}$ hourly concentrations (in $\mu\mathrm{g\,m^{-3}}$) using SCHULTE (Case-1) in red line and SIRANE (Case-7) in blue line."

4. Figure 5 has too small fonts in x- and y-axis for it to be readable with 100% size. Maybe removing daily averaged concentration information from the y-axis as this becomes evident from the figure text? Same applies to Figures 6, 7, 8, 9, 10, 11, 12 and 13.

**Our response**:

The font size in Figures 5 to 13 has been changed.

5. There are still multiple challenges with the language and it should be carefully checked by e.g. a native speaker or language services.

50 **Our response**:

The English language has been revised.